# How Transformers Utilize Multi-Head Attention in In-Context Learning?
# A Case Study on Sparse Linear Regression

**Xingwu Chen**[*]
The University of Hong Kong
xingwu@connect.hku.hk

**Lei Zhao**[*]
University of Pennsylvania
leizhao7@wharton.upenn.edu

**Difan Zou**
The University of Hong Kong
dzou@cs.hku.hk

## Abstract

Despite the remarkable success of transformer-based models in various real-world tasks, their underlying mechanisms remain poorly understood. Recent studies have suggested that transformers can implement gradient descent as an in-context learner for linear regression problems and have developed various theoretical analyses accordingly. However, these works mostly focus on the expressive power of transformers by designing specific parameter constructions, lacking a comprehensive understanding of their inherent working mechanisms post-training. In this study, we consider a sparse linear regression problem and investigate how a trained multi-head transformer performs in-context learning. We experimentally discover that the utilization of multi-heads exhibits different patterns across layers: multiple heads are utilized and essential in the first layer, while usually one single head is dominantly utilized for subsequent layers. We provide a theoretical rationale for this observation: the first layer undertakes data preprocessing on the context examples, and the following layers execute simple optimization steps based on the preprocessed context. Moreover, we prove that such a preprocess-then-optimize algorithm can outperform naive gradient descent and ridge regression algorithms, which is also supported by our further experiments. Our findings offer insights into the benefits of multi-head attention and contribute to understanding the more intricate mechanisms hidden within trained transformers.

## 1 Introduction

Transformers [45] have emerged as a dominant force in machine learning, particularly in natural language processing. Transformer-based large language models such as Llama [42, 43] and the GPT family [36, 2, 12, 38], equipped with multiple heads and layers, showcasing exceptional learning and reasoning capabilities. One of the fundamental capabilities is in-context learning [12, 52], i.e., transformer can solve new tasks after prompting with a few context examples, without any further parameter training. Understanding their working mechanisms and developing reasonable theoretical explanations for their performance is vital and has gathered considerable research attention.

Numerous studies have been conducted to explore the expressive power of transformers, aiming to showcase their ability to tackle challenging tasks related to memorization [33], reasoning [24, 8,

---

[*]Equal contribution.

38th Conference on Neural Information Processing Systems (NeurIPS 2024).

28, 14], function approximation [26, 39], causal relationship [35], and simulating complex circuits [21, 31]. These endeavors typically aim to enhance our understanding of the capabilities and limitations of transformers when configured with varying numbers of heads[16] and layers. However, it's important to note that the findings regarding expressive power and complexity may not directly translate into explanations or insights into the behavior of trained transformer models in practical applications.

To perform a deeper understanding of the learning ability of transformer, a line of recent studies has been made to study the in-context learning performance of transformer by connecting it to certain iterative optimization algorithms [17, 3, 47]. These investigations have primarily focused on linear regression tasks with a Gaussian prior, demonstrating that a transformer with $L$ layers can mimic $L$ steps of gradient descent on the loss defined by contextual examples both theoretically and empirically[3, 53]. These observations have immediately triggered a series of further theoretical research, revealing that multi-layer and multi-head transformers can emulate a broad range of algorithms, including proximal gradient descent [8], preconditioned gradient descent [3, 50], functional gradient descent [15], Newton methods [22, 19], and ridge regression [8, 4]. However, these theoretical works are mostly built by designing specific parameter constructions, which may not reflect the key mechanism of trained transformers in practice. The precise roles of different transformer modules, especially for the various attention layers and heads, remain largely opaque, even within the context of linear regression tasks.

To this end, we take a deeper exploration regarding the working mechanism of transformer by investigating how transformers utilize multi-heads, at different layers, to perform the in-context learning. In particular, we consider the sparse linear regression task, i.e., the data is generated from a noisy linear model with sparse ground truth $\mathbf{w}^* \in \mathbb{R}^d$ with $\|\mathbf{w}^*\|_0 \le s \ll d$, and train a transformer model with multiple layers and heads. While a line of works also investigates this problem [20, 8, 1], understanding the key mechanisms behind trained transformers always requires more experimental and theoretical insights. Consequently, we empirically assess the importance of different heads at varying layers by selectively masking individual heads and evaluating the resulting performance degradation. Surprisingly, our observations reveal distinct utilization patterns of multi-head attention across layers of a trained transformer: *in the first attention layer, all heads appear to be significant; in the subsequent layers, only one head appears to be significant.* This phenomenon suggests that (1) employing multiple heads, particularly in the first layer, plays a crucial role in enhancing in-context learning performance; and (2) the working mechanisms of the transformer may be different for the first and subsequent layers.

Based on the experimental findings, we conjecture that muti-layer transformer may exhibit a preprocess-then-optimize algorithm on the context examples. Specifically, transformers utilize all heads in the initial layer for data preprocessing and subsequently employ a single head in subsequent layers to execute simple iterative optimization algorithms, such as gradient descent, on the preprocessed data. We then develop the theory to demonstrate that such an algorithm can be indeed implemented by a transformer with multiple heads in the first layer and one head in the remaining layers, and can achieve substantially lower excess risk than gradient descent and ridge regression (without data preprocessing). The main contributions of this paper are highlighted as follows:

- We empirically investigate the role of different heads within transformers in performing in-context learning. We train a transformer model based on the data points generated by the noisy sparse linear model. Then, we reveal a distinct utilization pattern of multi-head attention across layers: while the first attention layer tended to evenly utilize all heads, subsequent layers predominantly relied on a single head. This observation suggests that the working mechanisms of multi-head transformers may vary between the first and subsequent layers.

- Building upon our empirical findings, we proposed a possible working mechanism for multi-head transformers. Specifically, we hypothesized that transformers use the first layer for data preprocessing on in-context examples, followed by subsequent layers performing iterative optimizations on the preprocessed data. To substantiate this hypothesis, we theoretically demonstrated that, by constructing a transformer with mild size, such a preprocess-then-optimize algorithm can be implemented using multiple heads in the first layers and a single head in subsequent layers.

- We further validated our proposed mechanism by comparing the performance of the preprocess-then-optimize algorithm with multi-step gradient descent and ridge regression solution, which can be implemented by the single-head transformers. We prove that the preprocess-then-optimize algorithm can achieve lower excess risk compared to these traditional methods, which is also verified by our numerical experiments. This aligns with our empirical findings, which indicated that multi-head transformers outperformed ridge regression in terms of excess risk.

- To further validate our theoretical framework, we conducted additional experiments. Specifically, we performed probing on the output of the first layer of the transformer and demonstrated that representations generated by transformers with more heads led to lower excess risk after gradient descent. These experiments provided further support for our explanation on the working mechanism of transformers.

## 2 Preliminaries

**Sparse Linear Regression.** We consider sparse linear models where $(\mathbf{x}, y) \sim \mathsf{P} = \mathsf{P}^{\text{lin}}_{\mathbf{w}^\star}$ is sampled as $\mathbf{x} \sim \mathsf{N}(\mathbf{0}, \boldsymbol{\Sigma})$, $y = \langle \mathbf{w}^\star, \mathbf{x} \rangle + \mathsf{N}(0, \sigma^2)$, where the $\boldsymbol{\Sigma}$ is a diagonal matrix and ground truth $\mathbf{w}^\star \in \mathbb{R}^d$ satisfies $\|\mathbf{w}^\star\|_0 \leq s$. Then, we define the population risk of a parameter $\mathbf{w}$ as follows:

$$L(\mathbf{w}) := \mathbb{E}_{(\mathbf{x}, y) \sim \mathsf{P}}\big[(\langle \mathbf{x}, \mathbf{w} \rangle - y)^2\big].$$

Moreover, we are interested in the excess risk, i.e., the gap between the population risk achieved by $\mathbf{w}$ and the optimal one:

$$\mathcal{E}(\mathbf{w}) := L(\mathbf{w}) - \min_{\mathbf{w}} L(\mathbf{w}).$$

**Multi-head Transformers.** Transformers are a type of neural network with stacked attention and multi-layer perceptron (MLP) blocks. In each layer, the transformer first utilizes multi-head attention Attn to process the input sequence (or hidden states) $\mathbf{H} = [\mathbf{h}_1, \mathbf{h}_2, \ldots, \mathbf{h}_m] \in \mathbb{R}^{d_{\text{hid}} \times m}$. It computes $h$ different queries, keys, and values, and then concatenates the output of each head:

$$\mathsf{Attn}(\mathbf{H}, \theta_1) = \mathbf{H} + \mathsf{Concat}[\mathbf{V}_1 \mathsf{sfmx}(\mathbf{K}_1^\top \mathbf{Q}_1), \cdots, \mathbf{V}_h \mathsf{sfmx}(\mathbf{K}_h^\top \mathbf{Q}_h)],$$

where $\mathbf{V}_i = \mathbf{W}_{V_i} \mathbf{H}$, $\mathbf{Q}_i = \mathbf{W}_{Q_i} \mathbf{H}$, $\mathbf{V}_i = \mathbf{W}_{V_i} \mathbf{H}$ and $\theta_1 = \big\{ \mathbf{W}_{V_i}, \mathbf{W}_{K_i}, \mathbf{W}_{Q_i} \in \mathbb{R}^{d_{\text{hid}}/h \times d_{\text{hid}}} \big\}_{i=1}^h$ are learnable parameters. The MLP then applies a nonlinear element-wise operation:

$$\mathsf{MLP}(\mathbf{H}, \theta_2) = \mathbf{W}_1 \mathsf{ReLU}(\mathbf{W}_2 \mathsf{Attn}(\mathbf{H}, \theta_1)), \tag{2.1}$$

where $\theta_2 = \{\mathbf{W}_1, \mathbf{W}_2\}$ denotes the parameters of MLP. We remark that here some modules, such as layernorm and bias, are ignored for simplicity.

**Linear Attention-only Transformers** To perform an tractable theoretical investigation on the role of multi-head in the attention layer, we make further simplification on the transformer model by considering linear attention-only transformers. These simplifications are widely adopted in many recent works to study the behavior of transformer models [47, 53, 32, 3]. In particular, the $i$-th layer $\mathsf{TF}_i$ performs the following update on the input sequence (or hidden state) $\mathbf{H}^{(i-1)}$ as follows:

$$\mathbf{H}^{(i)} = \mathsf{TF}_i(\mathbf{H}^{(i-1)}) = \mathbf{W}_1\big(\mathbf{H}^{(i-1)} + \mathsf{Concat}[\{\mathbf{V}_i \mathbf{M} \mathbf{K}_i^\top \mathbf{Q}_i\}_{i=1}^h]\big), \quad \mathbf{M} := \begin{pmatrix} \mathbf{I}_n & \mathbf{0} \\ \mathbf{0} & \mathbf{0} \end{pmatrix} \in \mathbb{R}^{m \times m}, \tag{2.2}$$

where $\{\mathbf{W}_{V_i}, \mathbf{W}_{K_i}, \mathbf{W}_{Q_i} \in \mathbb{R}^{\frac{d_{\text{hid}}}{h} \times d_{\text{hid}}}\}_{i=1}^h$ and $\mathbf{W}_1 \in \mathbb{R}^{d_{\text{hid}} \times d_{\text{hid}}}$ are learnable parameters, note that as we ignore the ReLU activation in Eq.(2.1), so we merge the parameter $\mathbf{W}_1$ and $\mathbf{W}_2$ into one matrix $\mathbf{W}_1$. Besides, the mask matrix $\mathbf{M}$ is included in the attention to constrain the model focus the first $n$ in-context examples rather than the subsequent $m - n$ queries [3, 32, 54]. To adapt the transformer for solving sparse linear regression problems, we introduce additional linear layers $\mathbf{W}_E \in \mathbb{R}^{(d+1) \times d_{\text{hid}}}$ and $\mathbf{W}_O \in \mathbb{R}^{d_{\text{hid}} \times 1}$ for input embedding and output projection, respectively. Mathematically, let $\mathbf{E}$ denotes the input sequences with $n$ in-context example followed by $q$ queries,

$$\mathbf{E} = \begin{pmatrix} \mathbf{x}_1 & \mathbf{x}_2 & \cdots & \mathbf{x}_n & \mathbf{x}_{n+1} & \cdots & \mathbf{x}_{n+q} \\ y_1 & y_2 & \cdots & y_n & 0 & \cdots & 0 \end{pmatrix}. \tag{2.3}$$

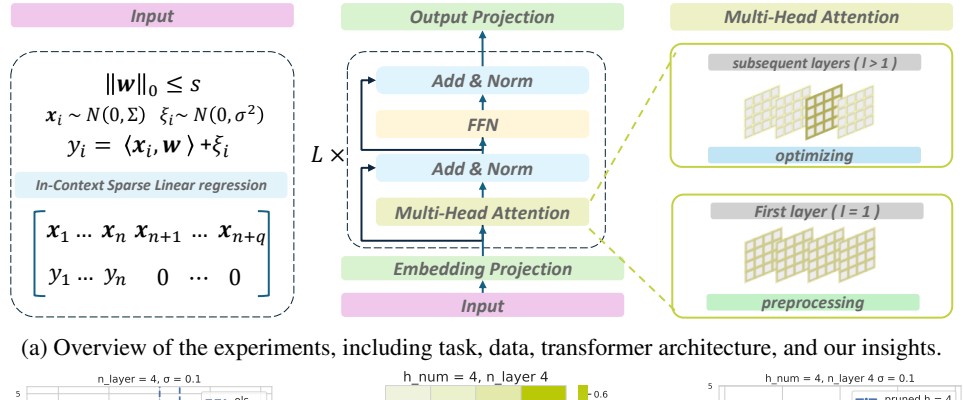

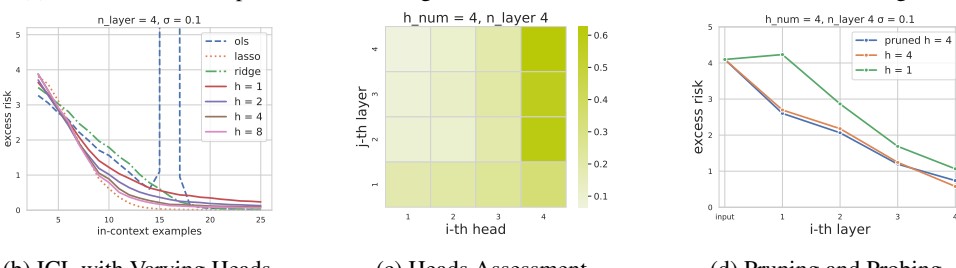

(a) Overview of the experiments, including task, data, transformer architecture, and our insights.

(b) ICL with Varying Heads      (c) Heads Assessment      (d) Pruning and Probing

Figure 1: Experimental Insights into Multi-head Attention for In-context Learning

Then model processes the input sequence $\mathbf{E}$, resulting in the output $\widehat{\mathbf{y}} \in \mathbb{R}^{1 \times (n+q)}$:

$$\widehat{\mathbf{y}} = \mathbf{W}_O \circ \mathsf{TF}_L \circ \cdots \circ \mathsf{TF}_1 \circ \mathbf{W}_E(\mathbf{E}),$$

here, $L$ is the layer number of the transformer, and $\widehat{y}_{i+n}$ is the prediction value for the query $\mathbf{x}_{i+n}$. During training, we set $q > 1$ for efficiency, and for inference and theoretical analysis, we set $q = 1$ and define the in-context learning excess risk $\mathcal{E}_{\mathsf{ICL}}$ as:

$$\mathcal{E}_{\mathsf{ICL}} := \mathbb{E}_{(\mathbf{x},y) \sim \mathsf{P}}(\widehat{y}_{n+1} - y_{n+1})^2 - \sigma^2.$$

**Notations.** For two functions $f(x) \geq 0$ and $g(x) \geq 0$ defined on the positive real numbers ($x > 0$), we write $f(x) \lesssim g(x)$ if there exists two constants $c, x_0 > 0$ such that $\forall x \geq x_0$, $f(x) \leq c \cdot g(x)$; we write $f(x) \gtrsim g(x)$ if $g(x) \lesssim f(x)$; we write $f(x) \asymp g(x)$ if $f(x) \lesssim g(x)$ and $g(x) \lesssim f(x)$. If $f(x) \lesssim g(x)$, we can write $f(x)$ as $O(g(x))$. We can also write write $f(x)$ as $\widetilde{O}(g(x))$ if there exists a constant $k > 0$ such that $f(x) \lesssim g(x) \log^k(x)$.

## 3 Experimental Insights into Multi-head Attention for In-context Learning

While previous work has demonstrated the in-context learning ability for sparse linear regression [20, 8], the hidden mechanism behind the trained transformer for solving this problem remains unclear. To this end, we design a series of experiments, utilizing techniques like probing [5] and pruning [27] to help us gain initial insights into how the trained transformer utilizes multi-head attention for this problem. For all experiments in Sections 3 and 6, we choose an encoder-based architecture as the backbone (see Figure 1a), set the hidden dimension $d_{\mathsf{hid}}$ to 256, and use the input sequence format shown in Eq.(2.3), where $d = 16$, $s = 4$, $\mathbf{x} \sim \mathsf{N}(\mathbf{0}, \mathbf{I})$, with varying noise levels, layers, and heads, Additional experimental details can be found in Appendix B. The experiments we designed are as follows:

**ICL with Varying Heads:** First, based on the experiment results by [8], we further investigate the performance of transformers in solving the in-context sparse linear regression problem with varying attention heads. An example can be found in Figure 1b, where we display the excess risk for different models when using different numbers of in-context examples. We can observe that given few-shot in-context examples, transformers can outperform OLS and ridge. Moreover, we can also clearly observe the benefit of using multiple heads, which leads to lower excess risk when increasing

the number of heads. This **highlights the importance of multi-head attention in transformer to perform in-context learning**.

**Heads Assessment:** Based on Eq.(2.2), we know that the $j$-th head at the $i$-th layer corresponds to the subspace of the intermediate output from $(j-1) \cdot d_{\mathsf{hid}}/h$ to $j \cdot d_{\mathsf{hid}}/h - 1$. To assess the importance of each attention head, we can mask the particular head by zeroing out the corresponding output entries, while keeping other dimensions unchanged. Then, let $(i, j)$ be the layer and head indices, we evaluate the risk change before and after head masking, denoted by $\Delta \mathcal{E}_{\mathsf{ICL}(i,j)}$. Then we normalize the risk changes in the same layer to evaluate their relative importance:

$$\mathcal{W}_{i,j} = \frac{\Delta \mathcal{E}_{\mathsf{ICL}(i,j)}}{\sum_{k=1}^{h} \Delta \mathcal{E}_{\mathsf{ICL}(i,k)}}. \tag{3.1}$$

An example can be found in Figure 1c. We can observe that in the first layer, no head distinctly outweighs the others, while in the subsequent layers, there always exists a head that exhibits higher importance than others. This gives us insight that **in the first attention layer, all heads appear to be significant, while in the subsequent layers, only one head appears to be significant**.

**Pruning and Probing:** To further validate our findings in the previous experiments, we prune the trained model by (1) retaining all heads in the first layer; and (2) only keeping the most important head and zeroing out others for the subsequent layers. Then the pruned model, referred to as the "pruned transformer", will be fine-tuned with with the same training data. We then use linear probes [6] to evaluate the prediction performance for different layers. An example can be found in Figure 1d, we can find that the "pruned transformer" and the original model exhibit almost the same performance for each layer. Additionally, compared to the model with single-head attention, we observe that the probing result is largely different between single-head transformers and the "pruned transformers", the latter has better performances compared to the former. Noting that the main difference between them is the number of heads in the first layer (subsequent layers have the same structure), it can be deduced that **the working mechanisms of the multi-head transformer may be different for the first and subsequent layers**.

## 4 Potential Mechanism Behind Trained transformer

Based on the experimental insights from Section 3, we found that all heads in the first layer of the trained transformer are crucial, while in subsequent layers, only one head plays a significant role. Furthermore, by checking the result for probing and pruning, we can find that the working mechanisms of the transformer may be different for the first and subsequent layers. To this end, we hypothesize that the multi-layer transformer may implement a preprocess-then-optimize to perform the in-context learning, i.e., the transformer first performs preprocessing on the in-context examples using the first layer and then implements multi-step iterative optimization algorithms on the preprocessed in-context examples using the subsequent layers.

We note that [24] adapts a similar two-phase idea to explain how transformer learning specific functions in context, in their constructed transformers, the first few layers utilize MLPs to compute an appropriate representation for each entry, while the subsequent layers utilize the attention module to implement gradient descent over the context. We highlight that our algorithm mainly focus on utilizing multihead attention, and it aligns well with the our experimental observation and intuition. The details of our algorithm are as follows:

### 4.1 Preprocessing on In-context Examples

First, as the multihead attention is designed to facilitate to model to capture features from different representation subspaces [45], we abstract the algorithm implementation by the first layer of the transformers as a preprocessing procedure. In general, for the sparse linear regression, a possible data preprocessing method is to perform reweighting of the data features by emphasizing the features that correspond to the nonzero entries of the ground truth $\mathbf{w}^*$ and disregard the remaining features. In the idealized case, if we know the nonzero support of $\mathbf{w}^*$, we can trivially zero out the date features of $\mathbf{x}$ on the complement of the nonzero support, as a data preprocessing procedure, and perform projected gradient descent to obtain the optimal solution.

In general, the nonzero support of $\mathbf{w}^*$ is intractable to the learner, so that one cannot perform idealized masking-related data preprocessing. However, one can still perform estimations on the importance of data features by examining their correlation with the target. In particular, note that we have $y = \langle \mathbf{w}^*, \mathbf{x} \rangle + \xi_i = \sum_{i=1}^{d} w_i^* x_i + \xi_i$, implying that $r_i := \mathbb{E}[x_i y] = \mathbb{E}[\sum_{i=1}^{d} w_i^* x_i \cdot x_i] + \mathbb{E}[\xi x_i] = w_i^* \mathbb{E}[x_i^2]$ if considering independent data features. Then it is clear that such a correlation between the feature and label will be nonzero only when $|w_i^*| \neq 0$. Therefore, instead of knowing the nonzero support of $\mathbf{w}^*$, we can instead calculate such a correlation to perform reweighting on the data features. Noting that the transformer is provided with $n$ in-context examples $\{(\mathbf{x}_i, y_i)\}_{i=1}^{n}$, such correlations can be estimated accordingly: $\widehat{r}_j = \frac{1}{n} \sum_{i=1}^{n} x_{ij} y_i$, which will be further used to perform the data preprocessing on the in-context examples. We summarize this procedure in Alg. 1.

---

**Algorithm 1** Data preprocessing for in-context examples

---

1: **Input :** Sequence with $\{(\mathbf{x}_i, y_i)\}_{i=1}^{n}, \{(\mathbf{x}_i, 0)\}_{i=n+1}^{n+q}$ as in-context examples/queries.
2: **for** $k = 1, \ldots, n$ **do**
3:     Compute $\widetilde{\mathbf{x}}_k$ by $\widetilde{\mathbf{x}}_k = \widehat{\mathbf{R}} \mathbf{x}_k$, where $\widehat{\mathbf{R}} = \text{diag}\{\widehat{r}_1, \widehat{r}_2, \ldots, \widehat{r}_d\}$, where $\widehat{r}_j$ is given by

$$\widehat{r}_j = \frac{1}{n} \sum_{i=1}^{n} x_{ij} y_i. \tag{4.1}$$

4: **end for**
5: **Output :** Sequence with the preprocessed in-context examples/queries $\{(\widetilde{\mathbf{x}}_i, y_i)\}_{i=1}^{n}, \{(\widetilde{\mathbf{x}}_i, 0)\}_{i=n+1}^{n+q}$.

---

The preprocessing procedure aligns well with the structure of a multi-head attention layer with linear attention, which motivates our theoretical construction of the desired transformer. In particular, each head of the attention layer can be conceptualized as executing specific operations on a distinct subset of data entries. Then, the linear query-key calculation, represented as $(\mathbf{W}_{K_i} \mathbf{H})^\top \mathbf{W}_{Q_i} \mathbf{H}$, where $\mathbf{H} = \mathbf{E}$ denotes the input sequence embedding matrix, effectively estimates correlations between the $i$-th subset of data entries and the corresponding label $y_i$. Here, $\mathbf{W}_{K_i}$ and $\mathbf{W}_{Q_i}$ selectively extract entries from the $i$-th subset of features and the label, respectively, akin to an "entries selection" process. Furthermore, when combined with the value calculation $\mathbf{W}_{V_i} \mathbf{H}$, each head of the attention layer conducts correlation calculations for the $i$-th subset of features and subsequently employs them to reweight the original features within the same subset. Consequently, by stacking the outputs of multiple heads, all data features can be reweighted accordingly, which matches the design of the proposed preprocessing procedure in Alg. 1. We formally prove this in the following theorem.

**Proposition 4.1** (Single-layer multi-head transformer implements Alg. 1). *There exists a single-layer transformer function* $\mathsf{TF}_1$, *with $d$ heads and $d_{\text{hid}} = 3d$ hidden dimension, together with an input embedding layer with weight* $\mathbf{W}_E \in \mathbb{R}^{d_{\text{hid}} \times d}$, *that can implement Alg. 1. Let $\mathbf{E}$ be the input sequence defined in Eq.(2.3) and $\widetilde{\mathbf{x}}_i = \widehat{\mathbf{R}} \mathbf{x}$ be the preprocessed features defined in Alg. 1, it holds that*

$$\mathbf{H}^{(1)} := \mathsf{TF}_1 \circ \mathbf{W}_E(\mathbf{E}) = \begin{pmatrix} \widetilde{\mathbf{x}}_1 & \widetilde{\mathbf{x}}_2 & \cdots & \widetilde{\mathbf{x}}_n & \widetilde{\mathbf{x}}_{n+1} & \cdots & \widetilde{\mathbf{x}}_{n+q} \\ y_1 & y_2 & \cdots & y_n & 0 & \cdots & 0 \\ \vdots & \vdots & \ddots & \vdots & \vdots & \ddots & \vdots \end{pmatrix}, \tag{4.2}$$

*where $\cdots$ in third row implies arbitrary values.*

## 4.2 Optimizing Over Preprocessed In-Context Examples

Based on the experimental results, we observe that the subsequent layers of transformers dominantly rely on one single head, suggesting their different but potentially simpler behavior compared to the first one. Motivated by a series of recent work [47, 15, 53, 3] that reveal the connection between gradient descent steps and multi-layer single-head transformer in the in-context learning tasks, we conjecture that the subsequent layers also implement iterative optimization algorithms such as gradient descent on the (preprocessed) in-context examples.

To maintain clarity in our construction and explanation, in each layer, we use a linear projection $\mathbf{W}_1^{(i)}$ to rearrange the dimensions of the sequence processed by the multi-head attention, resulting in the hidden state $\mathbf{H}^{(i)}$ of each layer. We refer to the first $d$ rows of the input data as $\mathbf{x}$, and the $(d+1)$-th row as the corresponding $y$. For example, in Eq.(4.2), we take the first $d$ rows, together with the

$(d + 1)$-th row, as the input data entry $\{\widetilde{\mathbf{x}}_i, y_i\}_{i=1}^{n+1}$. Then, the following proposition shows that the subsequent layers of transformer can implement multi-step gradient descent on the preprocessed in-context examples $\{(\widetilde{\mathbf{x}}_i, y_i)\}_{i=1,\dots,n}$.

**Proposition 4.2** (Subsequent single-head transformer implements multi-step GD). *There exists a transformer with $k$ layers, 1 head, $d_{\mathsf{hid}} = 3d$, let $\widehat{y}_{n+1}^\ell$ be the prediction representation of the $\ell$-th layer, then it holds that $\widehat{y}_{(n+1)}^\ell = \langle \mathbf{w}_{\mathsf{gd}}^\ell, \widetilde{\mathbf{x}}_{n+1} \rangle$, where $\widetilde{\mathbf{x}}_{n+1} = \widehat{\mathbf{R}}\mathbf{x}_{n+1}$ denotes the preprocessed data feature, $\mathbf{w}_{\mathsf{gd}}^\ell$ is defined as $\mathbf{w}_{\mathsf{gd}}^0 = 0$ and as follows for $\ell = 0, \dots, k-1$:*

$$\mathbf{w}_{\mathsf{gd}}^{\ell+1} = \mathbf{w}_{\mathsf{gd}}^\ell - \eta \nabla \widetilde{L}(\mathbf{w}_{\mathsf{gd}}^\ell), \quad \text{where} \quad \widetilde{L}(\mathbf{w}) = \frac{1}{2n}\sum_{i=1}^n (y_i - \langle \mathbf{w}, \widetilde{\mathbf{x}}_i \rangle)^2. \tag{4.3}$$

The proof of Propositions 4.1 and 4.2 can be found in Appendix C, combining these two propositions , we show that the multi-layer transformer with multiple heads in the first layer and one head in the subsequent layers can implement the proposed preprocess-then-optimization algorithm. In the next section, we will establish theories to demonstrate that such an algorithm can indeed achieve smaller excess risk than standard gradient descent and ridge regression solutions of the sparse linear regression problem.

# 5   Excess Risk of the Preprocess-then-optimize Algorithm

In this section, we will develop the theory to demonstrate the improved performance of the preprocess-then-optimize algorithm compared to the gradient descent algorithm on the raw inputs. The proof for Theorem 5.1, 5.2, and 5.3 can be found in Appendix D, E, and F, respectively.

We first denote $\widetilde{\mathbf{w}}_{\mathsf{gd}}^t$ as the estimator obtained by $t$-step GD on $\{(\widetilde{\mathbf{x}}_i, y_i)\}_{i=1}^n$, which can be viewed as the solution generated by the $t + 1$-layer transformer based on our discussion in Section 4, and $\mathbf{w}_{\mathsf{gd}}^t$ as the estimator obtained by $t$-step GD on $\{(\mathbf{x}_i, y_i)\}_{i=1}^n$. Before presenting our main theorem, we first need to redefine the excess risk of GD on $\{(\widetilde{\mathbf{x}}_i, y_i)\}_{i=1}^n$. Note that in our algorithm, the learned predictor takes the form $\mathbf{x} \to \langle \widehat{\mathbf{R}}\mathbf{x}, \widetilde{\mathbf{w}}_{\mathsf{gd}}^t \rangle$. Consequently, the population risk of a parameter $\widetilde{\mathbf{w}}_{\mathsf{gd}}^t$ is naturally defined as $\widetilde{L}(\widetilde{\mathbf{w}}_{\mathsf{gd}}^t) := \frac{1}{2} \cdot \mathbb{E}_{(\mathbf{x},y)\sim\mathsf{P}}\left[(\langle \widehat{\mathbf{R}}\mathbf{x}, \widetilde{\mathbf{w}}_{\mathsf{gd}}^t \rangle - y)^2\right]$, and the excess risk is then defined as $\mathcal{E}(\mathbf{w}) := \widetilde{L}(\mathbf{w}) - \min_{\mathbf{w}} \widetilde{L}(\mathbf{w})$ [2]. Next, we provide the upper bound of the excess risk for $\mathcal{E}(\widetilde{\mathbf{w}}_{\mathsf{gd}}^t)$ and $\mathcal{E}(\mathbf{w}_{\mathsf{gd}}^t)$ respectively.

**Theorem 5.1.** *Denote $\mathcal{S} := \{i : w_i^\star \neq 0\}$ and $\mathbf{R} = \mathsf{diag}\{r_1, \dots, r_d\}$, where $r_j = \sum_{i=1}^d w_i^\star \Sigma_{ij}$. Suppose that there exist a $\beta > 0$ such that $\min_{i\in\mathcal{S}} |r_i| \geq \beta$, $\|\mathbf{R}\|_2, \|\boldsymbol{\Sigma}\|_2, \|\mathbf{w}^\star\|_2 \simeq O(1)$ and $n \gtrsim 1/\beta^2 \cdot t^2 s \cdot \left(\mathrm{Tr}^{2/3}(\boldsymbol{\Sigma}) + \mathrm{Tr}(\mathbf{R}\boldsymbol{\Sigma}\mathbf{R})\right) \cdot \mathsf{poly}(\log(d/\delta))$. Then set $\eta \lesssim 1/\|\mathbf{R}\boldsymbol{\Sigma}\mathbf{R}\|_2$ and*

$$\eta t \simeq \frac{1}{\beta} \cdot \left(\frac{\sigma^2 \mathrm{Tr}(\mathbf{R}\boldsymbol{\Sigma}\mathbf{R})\log(d/\delta)}{n} + \frac{\sigma^2 s \mathrm{Tr}(\boldsymbol{\Sigma})\log^2(d/\delta)}{n^2}\right)^{-1/2},$$

*it holds that*

$$\mathcal{E}(\widetilde{\mathbf{w}}_{\mathsf{gd}}^t) \lesssim \frac{\log t}{\beta}\sqrt{\frac{\sigma^2 \mathrm{Tr}(\mathbf{R}\boldsymbol{\Sigma}\mathbf{R})\log(d/\delta)}{n} + \frac{\sigma^2 s \mathrm{Tr}(\boldsymbol{\Sigma})\log^2(d/\delta)}{n^2}},$$

*with probability at least $1 - \delta$.*

Theorem 5.1 provides an upper bound on the excess risk achieved by the preprocess-then-optimize algorithm, where we tuned learning rate $\eta$ to balance the bias and variance error. Then, it can be seen that the risk bound is valid if $\mathrm{Tr}(\mathbf{R}\boldsymbol{\Sigma}\mathbf{R})/n \to 0$ and $\mathrm{Tr}(\boldsymbol{\Sigma})s/n^2 \to 0$ when $n \to \infty$. This can be readily satisfied if we have $\|\mathbf{w}^*\|_2$ and $\mathrm{Tr}(\boldsymbol{\Sigma})$ be bounded by some reasonable quantities that are independent of the sample size $n$, which are the common assumptions made in many prior works [58, 57, 9]. Besides, it can be also seen that the excess risk bound explicitly depends on the sparsity parameter $s$ and lower sparsity implies better performance. This implies the ability of the proposed preprocess-then-optimize for discovering and leveraging the nice sparse structure of the ground truth.

---

[2]Here for the ease to presentation and comparison, we slightly abuse the notation of $\mathcal{E}(\mathbf{w})$ by extending it to $\widetilde{\mathbf{w}}_{\mathsf{gd}}^t$, although $\mathcal{E}(\mathbf{w})$ is originally defined for the estimator for the raw feature vector $\mathbf{x}$.

As a comparison, the following theorem states the excess risk bound for the standard gradient descents on the raw features. To make a fair comparison, we consider using the same number of steps but allow the step size to be tuned separately.

**Theorem 5.2.** *Suppose that* $\|\mathbf{\Sigma}\|, \|\mathbf{w}^\star\|_2 \simeq O(1)$ *and* $n \gtrsim t^2(\mathrm{Tr}(\mathbf{\Sigma}) + \log{(1/\delta)})$. *When* $\eta \lesssim$ $1/\|\mathbf{\Sigma}\|_2$ *and* $\eta t \simeq \left(\frac{\sigma^2 \mathrm{Tr}(\mathbf{\Sigma}) \log{(d/\delta)}}{n}\right)^{-1/2}$, *it holds that*

$$\mathcal{E}\left(\mathbf{w}_{\mathsf{gd}}^t\right) \lesssim \log t \cdot \sqrt{\frac{\sigma^2 \mathrm{Tr}(\mathbf{\Sigma}) \log{(d/\delta)}}{n}},$$

*with probability at least* $1 - \delta$.

We are now able to make a rough comparison between the excess risk bounds in Theorems 5.1 and 5.2. Then, it is clear that $\mathcal{E}(\widetilde{\mathbf{w}}_{\mathsf{gd}}^t) \lesssim \mathcal{E}(\mathbf{w}_{\mathsf{gd}}^t)$ requires $\mathrm{Tr}(\mathbf{R}\mathbf{\Sigma}\mathbf{R})/\beta^2 \lesssim \mathrm{Tr}(\mathbf{\Sigma})$ and $s/(n^2\beta^2) \leq 1/n$. Specifically, we can consider the case that $\mathbf{\Sigma}$ to be a diagonal matrix, assume $w_i^\star \sim \mathsf{U}\{-1/\sqrt{s}, 1/\sqrt{s}\}$ has a restricted uniform prior for $i \in \mathcal{S}$ and $\min_{i\in\mathcal{S}} \mathbf{\Sigma}_{ii} \geq 1/\kappa$ for some constant $\kappa > 1$, we can get $\beta \geq \sqrt{1/(s\kappa^2)}$, thus $\mathrm{Tr}(\mathbf{R}\mathbf{\Sigma}\mathbf{R})/\beta^2 \leq \kappa^2 \sum_{i:w_i^\star \neq 0} \mathbf{\Sigma}_{ii}$ and $s/(n^2\beta^2) \leq \kappa^2 s^2/n^2$. Note that $|\mathcal{S}| = s \ll d$, then if the covariance matrix $\mathbf{\Sigma}$ has a flat eigenspectrum such that $\sum_{i\in\mathcal{S}} \mathbf{\Sigma}_{ii} \ll \sum_{i\in[d]} \mathbf{\Sigma}_{ii} = \mathrm{Tr}(\mathbf{\Sigma})$, we have $\mathrm{Tr}(\mathbf{R}\mathbf{\Sigma}\mathbf{R})/\beta^2 \leq \mathrm{Tr}(\mathbf{\Sigma})$ and $s/(n^2\beta^2) \leq \kappa^2 s^2/n$ if $s = o\left(\min\{d, \sqrt{n}\}\right)$. This suggests that the preprocess-then-optimization algorithm can outperform the standard gradient descent for solving a sparse linear regression problem with $s = o\left(\min\{d, \sqrt{n}\}\right)$.

To make a more rigorous comparison, we next consider the example where $x_i \overset{\text{i.i.d.}}{\sim} \mathsf{N}(\mathbf{0}, \mathbf{I})$, based on which we can get the upper bound for our algorithm and the lower bound for OLS, ridge regression, and finite-step GD.

**Theorem 5.3.** *Suppose* $\mathcal{S}$ *with* $|\mathcal{S}| = s$ *is selected such that each element is chosen with equal probability from the set* $\{1, 2, \ldots, d\}$ *and* $w_i^\star \sim \mathsf{U}\{-1/\sqrt{s}, 1/\sqrt{s}\}$ *has a restricted uniform prior for* $i \in \mathcal{S}$, $\|\mathbf{w}^\star\|_2 \simeq \Theta(1)$ *and* $n \gtrsim t^2 s^3 d^{2/3}$. *Then there exists a choice of* $\eta$ *and* $t$ *such that*

$$\mathcal{E}\left(\widetilde{\mathbf{w}}_{\mathsf{gd}}^t\right) \lesssim \sigma^2 \log^2{\left(ns/\sigma^2\right)} \log^2{(d/\delta)} \cdot \left(\frac{s}{n} + \frac{ds^2}{n^2}\right),$$

*with probability at least* $1 - \delta$. *Besides, let* $\widehat{\mathbf{w}}_\lambda$ *be the ridge regression estimator with regularized parameter* $\lambda$, *and* $\mathbf{w}_{\mathrm{ols}}$ *be the OLS estimator, it holds that*

$$\mathbb{E}_{\mathbf{w}^\star}[\mathcal{E}(\mathbf{w})] \gtrsim \begin{cases} \frac{\sigma^2 d}{n} & n \gtrsim d + \log{(1/\delta)} \\ 1 - \frac{n}{d} + \frac{\sigma^2 n}{d} & d \gtrsim n + \log{(1/\delta)}, \end{cases}$$

*with probability at least* $1 - \delta$, *where* $\mathbf{w} \in \{\widehat{\mathbf{w}}_\lambda, \mathbf{w}_{\mathrm{ols}}, \mathbf{w}_{\mathsf{gd}}^t\}$.

It can be seen that for a wide range of under-parameterized and over-parameterized cases, $\widetilde{\mathbf{w}}_{\mathsf{gd}}^t$ has a smaller excess risk than ridge regression, standard gradient descent, and OLS. In particular, consider the setting $\sigma^2 = 1$, in the over-parameterized setting that $d \gtrsim n$, the excess risk bound of preprocess-then-optimize is $\widetilde{O}(ds/n^2)$, which also outperforms the $\widetilde{\Omega}(1)$ bound achieved by OLS, ridge regression, and standard gradient descent if the sparsity satisfies $s = O(n^2/d)$ (in fact, this condition can be certainly removed as $\mathcal{E}(\widetilde{\mathbf{w}}_{\mathsf{gd}}^t)$ also has a naive upper bound $\widetilde{O}(1)$). In the under-parameterized case that $d \lesssim n$, it can be readily verified that the data preprocessing can lead to a $\widetilde{O}(s/n)$ excess risk, which is strictly better than the $\widetilde{\Omega}(d/n)$ risk achieved by OLS, ridge regression, and standard gradient descent. Moreover, it is well known that Lasso can achieve $\widetilde{O}(s/n)$ excess risk bound in the setting of Theorem 5.3. Then, by comparing with our results, we can also conclude that the preprocess-then-optimize algorithm can be comparable to Lasso up to logarithmic factors when $d \lesssim n$, while becomes worse when $d \gtrsim n$.

## 6 Experiments

In Section 3, we conduct several experiments, and based on the observations, we propose that a trained transformer can apply a preprocess-then-optimize algorithm: (1) In the first layer, the transformer can apply a preprocessing algorithm (Alg. 1) on the in-context examples utilizing multi-head attention.

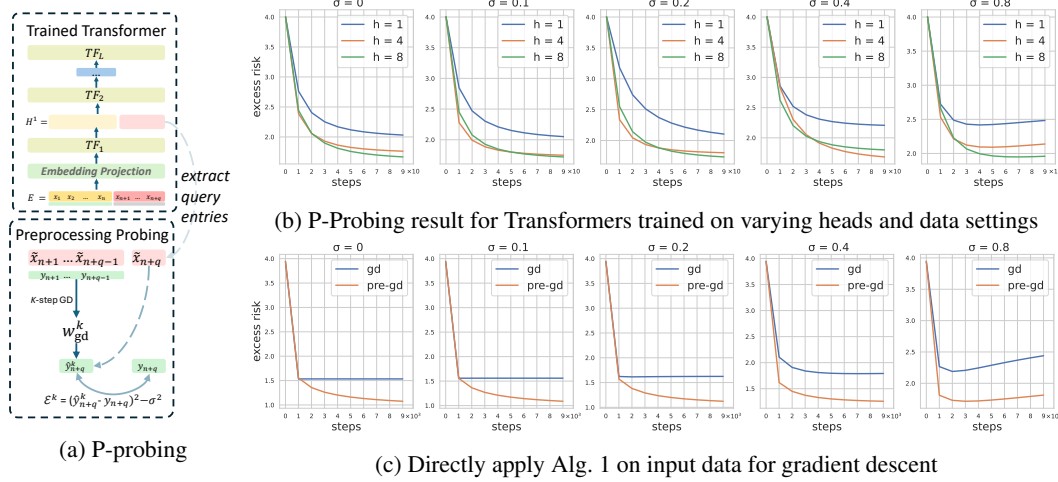

(a) P-probing

(b) P-Probing result for Transformers trained on varying heads and data settings

(c) Directly apply Alg. 1 on input data for gradient descent

Figure 2: Supporting experiments for our preprocess-then-optimize algorithm and theoretical analysis

(2) In the subsequent layers, the transformer applies a gradient descent algorithm on the preprocessed data utilizing single-head attention. While the second part is supported by extensive theoretical analysis and experimental evidence [47, 15, 53, 3], here we develop a technique called preprocessing probing (P-probing) on the trained transformer to support the first part of our algorithm. We also directly apply Alg. 1 on the in-context examples and then check the excess risk for multiple-step gradient descent to verify the effectiveness of our algorithm and theoretical analysis.

**P-probing:** To verify the existence of a preprocessing procedure in the trained transformer, we develop a "preprocessing probing" (P-probing) technique on the trained transformers, as illustrated in Figure 2a. For a trained transformer, we first set the input sequence as in Eq.(2.3), where the first $n$ examples $\{\mathbf{x}\}_{i=1}^n$ have the corresponding labels $\{y\}_{i=1}^n$, and the following $q$ query entries only have $\{\mathbf{x}_i\}_{i=n+1}^{n+q}$ in the sequence. Then, we extract the last $q$ vectors in the output hidden state $\mathbf{H}^1$ from the first layer of the transformer and treat these data as processed query entries. Next, we conduct gradient descent on the first $q-1$ query entries with their corresponding $y$, computing the excess risk on the last query. Additional experimental details can be found in Appendix B. We adapt this technique based on the intuition that, according to our theoretical analysis, we can extract the preprocessed entry $\{\widetilde{\mathbf{x}}_i\}_{i=n+1}^{n+q}$ from $\mathbf{H}^1$, besides, the excess risk computed by the preprocessed data has a better upper bound guarantee compared to raw data without preprocessing under the same number of gradient descent steps, so if the trained transformer utilize multihead attention for preprocess, compared with single head attention, the queries entries extract from $\mathbf{H}^1$ by multihead attention can have better gradient descent performance compared with single head attention.

**Verifying the benefit of preprocessing:** To further support the effectiveness of our algorithm, we directly apply Alg. 1 on the input data $\{\mathbf{x}_i, y_i\}_{i=1}^{n+1}$, and then implement gradient descent on the example entries $\{\mathbf{x}_i, y_i\}_{i=1}^n$ and compute the excess risk with the last query $\{\widehat{\mathbf{x}}_{n+1}, y_{n+1}\}$, we refer this procedure as `pre-gd`. We compare `pre-gd` with the excess risk obtained by directly applying gradient descent without preprocessing (referred to as `gd`). For all experiments (both P-probing and this), we set $\mathbf{w}_{\text{gd}}^0 = \mathbf{0}$ and tune the learning rate $\eta$ for each model by choosing from $[1, 10^{-1}, 10^{-2}, 10^{-3}, 10^{-4}, 10^{-5}, 10^{-6}]$ with the lowest average excess risk.

Based on Figure 2b, we can observe that compared to the transformer with single-head attention ($h = 1$), the query entries extracted from the transformer with multiple heads ($h = 4, 8$) preserve better convergence performance and can dive into a lower risk. This aligns well with our experiment result in Figure 2c, where compared to `gd`, the data preprocessed by Alg. 1 preserves better convergence performance and can dive into a lower risk space, supporting the existence of the preprocessing procedure in the trained transformer. Moreover, Figure 2c also aligns well with our theoretical analysis, where we provide a better upper bound for convergence guarantee for our algorithm compared to ridge regression and OLS.

# 7 Conclusions and Limitations

In this paper, we investigate a sparse linear regression problem and explore how a trained transformer leverages multi-head attention for in-context learning. Based on our empirical investigations, we propose a preprocess-then-optimize algorithm, where the trained transformer utilizes multi-head attention in the first layer for data preprocessing, and subsequent layers employ only a single head for optimization. We theoretically prove the effectiveness of our algorithm compared to OLS, ridge regression, and gradient descent, and provide additional experiments to support our findings.

While our findings provide promising insights into the hidden mechanisms of multi-head attention for in-context learning, there is still much to be explored. First, our work focuses on the case of sparse linear regression, and it may be beneficial to implement our experiment for more challenging or even real-world tasks. Additionally, as we adapt attention-only transformers for analysis simplification, the role of other modules, such as MLPs, are neglected. How these modules incorporate in real-world tasks remains unclear. Moreover, our analysis does not consider the training dynamics of transformers, while the theoretical analysis in [13] provides valuable insights into the convergence of single-layer transformers with multi-head attention, the training dynamics for multi-layer transformers remain unclear. How transformers learn to implement these algorithms is worth further investigation.

## Acknowledgements

We would like to thank the anonymous reviewers and area chairs for their helpful comments. This work is supported by NSFC 62306252, Guangdong NSF 2024A1515012444, Hong Kong ECS awards 27309624, and the central fund from HKU IDS.

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

## A  Additional Related Work

In addition to works towards understanding the expressive power of transformers that we introduced before, there is also a body of research on the mechanism interpretation and the training dynamics of transformers:

**Mechanism interpretation of trained transformers**  To understand the mechanisms in trained transformers, researchers have developed various techniques, including interpreting transformers into programming languages [18, 30, 48, 55], probing the behavior of individual layers [37, 51, 11, 7, 56], and incorporating transformers with other large language models to interpret individual neurons [10]. While these techniques provide high-level insights into transformer mechanism understanding, providing a clear algorithms behind the trained transformers is still very challenging.

**Training dynamics of transformers**  In parallel, a body of work has also investigated how transformers learn these algorithms, i.e., the training dynamics of transformers. Tarzanagh et al. [40] shows an equivalence between a single attention layer and a support vector machine. Zhang et al. [53], Ahn et al. [3] analyze the training dynamics of a single-head attention layer for in-context linear regression, where [53] demonstrates that it can converge to implement one-step gradient over in-context examples. Huang et al. [25], Chen et al. [13] extended these findings from linear attention to softmax settings, with [13] revealing that trained transformers with multi-head attention tend to utilize different heads for distinct tasks in various subspaces. Additionally, Tian et al. [41], Li et al. [29] study the convergence of transformers on sequences of discrete tokens. Gromov et al. [23], Men et al. [34] use experiments to show that in large language models, parameters in deeper layers are less critical compared to those in shallower layers. These works provide valuable insights towards the theoretical understanding of the training dynamics of transformers, which offer potential future extension aspects for our work.

## B  Additional Details for Sections 3 and 6

**Architecture and Optimization**  We conduct extensive experiments on encoder-only transformers with $d_{\text{hid}} = 256$, varying the number of heads $h \in \{1, 2, 4, 8\}$, layers $l \in \{3, 4, 5, 6\}$, and noise levels $\sigma \in \{0, 0.1, 0.2, 0.4, 0.8\}$. For the input sequence, we sample $\mathbf{x} \sim \mathsf{N}(\mathbf{0}, \mathbf{I})$. For $\mathbf{w}$, we first sample $\mathbf{w} \sim \mathsf{N}(\mathbf{0}, \mathbf{I}) \in \mathbb{R}^{16}$, and randomly choose $s = 4$ entries, setting the other elements to zero. Note that We don't apply positional encodings in our setting, as no positional information is needed in our input setting. To further support our preprocessing-then-optimize algorithm, we also try a decoder-only architecture(Figure 9), train models on other settings like stardand linear regression task $s = d = 16$ (Figure 10) and non-orthogonal data distributions (Figure 11) as comparisons in Appendix G. During training, we set $n = 12$ and $q = 4$, with a batch size of 64. We utilize the Adam optimizer with a learning rate $\gamma = 10^{-4}$ for 320000 updates. Each experiment takes about two hours on a single NVIDIA GeForce RTX 4090 GPU. We fix the random seed such that each model is trained and evaluated with the same training and evaluation dataset. We use HuggingFace [49] library to implement our models.

**ICL with Varying Heads**  We compare the model's performance with ridge regression, OLS, and lasso. For ridge regression and lasso, we tune $\lambda, \alpha \in \left\{1, 10^{-1}, 10^{-2}, 10^{-3}, 10^{-4}\right\}$ respectively for the lowest risk, as in [20].

From Figure 3, we can find that in most cases, transformers with single head ($h = 1$) exhibits higher risk compared to models with multiple heads ($h = 4, 8$). Note that in thesame subplot, models with different numbers of heads have the same number of parameters. This experiment highlights the importance of multi-head attention for transformers in in-context learning.

**Heads Assessment**  Here, we set $n = 10$ and $q = 1$, with an evaluation data size of 8192. For a model with $h$ heads and $l$ layers, we train $|\boldsymbol{\sigma}|$ models under different noise levels. We first compute the $\mathcal{W}^{h,l,\sigma}$ under different noise levels $\sigma$, then sort each row in $\mathcal{W}^{h,l,\sigma}$, and add them together as $\mathcal{W}^{h,l}_{\text{avg}} = \frac{1}{|\boldsymbol{\sigma}|} \sum_{\sigma \in \boldsymbol{\sigma}} \mathcal{W}^{h,l,\sigma}$, resulting in the final weight for each head. An example can be found in Fig 1c. In Fig 4, we present more results for different $h$ and $l$, and we also present the heat map for the decode-only transformers in Figure 9.

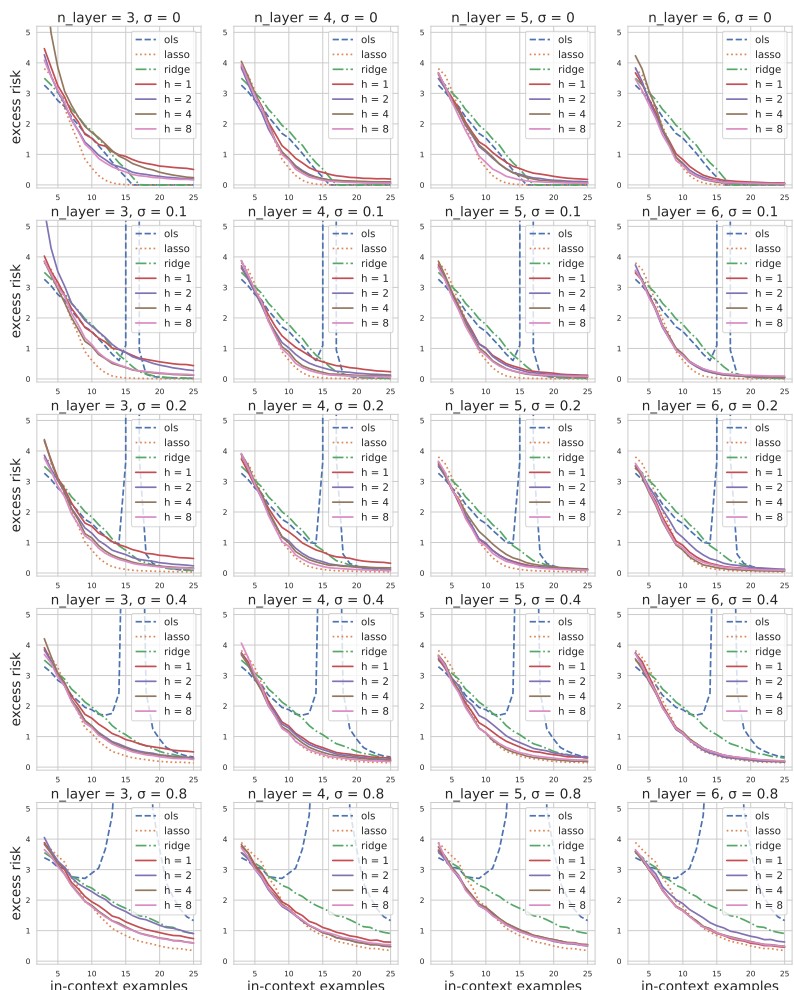

Figure 3: ICL with varying heads, layers and noise levels

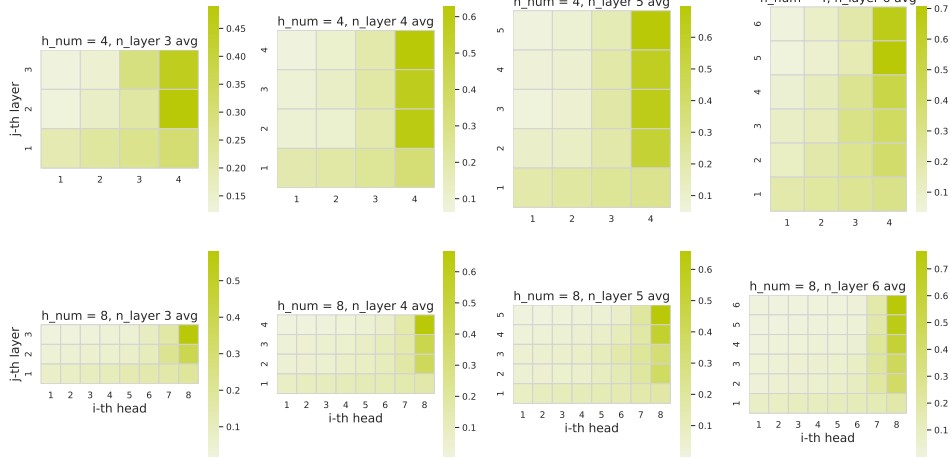

Figure 4: Head Assessment with varying heads, layers

From Fig 4, we can find that in most settings, each head contributes almost equally, while in the subsequent layers, there always exists a head that has a much larger weight than the others. This

indicates that in trained transformers for in-context learning, in the first attention layer, all heads appear to be significant, while in the subsequent layers, only one head appears to be significant.

**Pruning and Probing** Here, we also set $n = 10$ and $q = 1$, with an evaluation data size of $8192$. To further support our finding from the Head Assessment, we first prune the model based on our computed head weight $\mathcal{W}_{\text{avg}}^{h,l}$, where we keep all heads in the first layer, whereas we only keep the head with the highest score weight and mask the others. We then train the pruned model with the same method as before for $60000$ steps. In Fig 5, 6, 7, 8, we provide the Pruning and Probing results for different numbers of heads $h \in \{4, 8\}$ and noise levels $\sigma \in \{0, 0.1, 0.2, 0.4, 0.8\}$. It can be found that in almost all cases, the pruned model exhibits almost the same performance in each layer, while being largely different from the single-layer transformer. This further supports the results in the Heads Assessment and indicates that the working mechanisms of the multi-head transformer may be different for the first and subsequent layers.

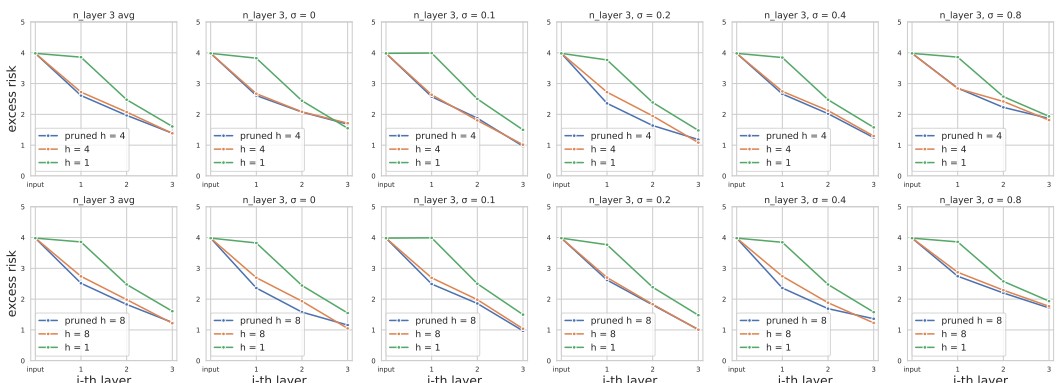

Figure 5: Pruning and Probing, 3 layers

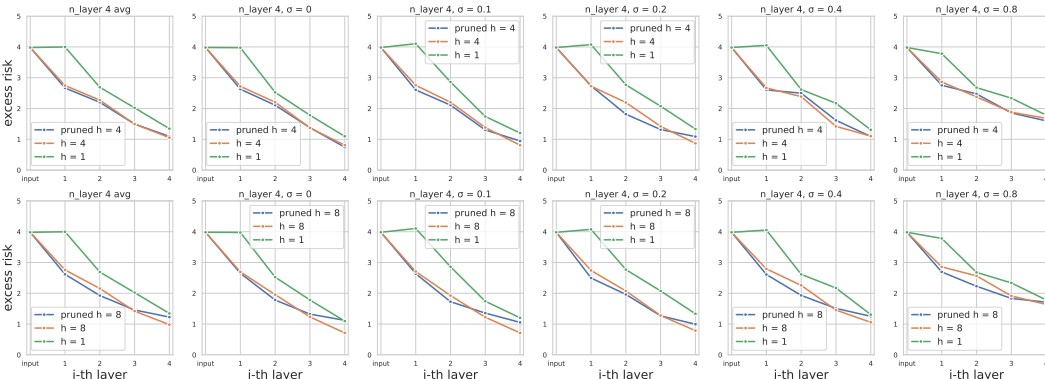

Figure 6: Pruning and Probing, 4 layers

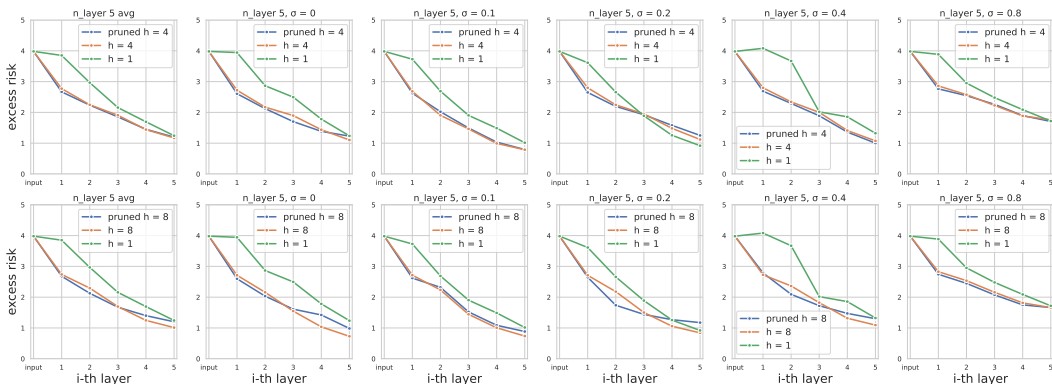

Figure 7: Pruning and Probing, 5 layers

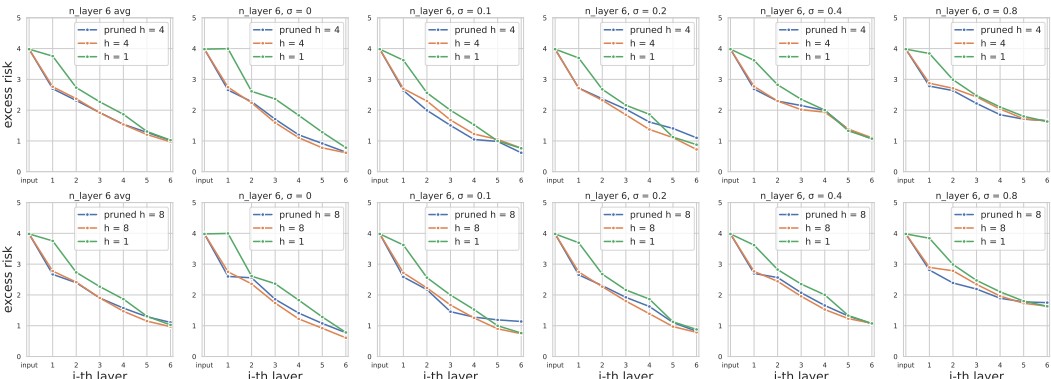

Figure 8: Pruning and Probing, 6 layers

**P-probing** Here, we also set $n = 117$ and $q = 11$, with an evaluation data size of 1024. We choose $n \gg q$ such that the model can handle more queries ($q = 11$) than those in the training ($q = 4$) process.

## C    Proof for Section 4

### C.1    Proof for Proposition 4.1

**Proposition C.1** (Restate of Proposition 4.1). *There exists a transformer with* 1 *layers,* $h = d$ *heads,* $d_{\mathsf{hid}} = 3d$ *and the input projection* $\mathbf{W}_E \in \mathbb{R}^{(d+1) \times d_{\mathsf{hid}}}$ *such that with the input sequence* $\mathbf{E}$ *set as Equation 2.3 the first attention layer can implement Algorithm 1 so that each of the enhanced data* $\{\widehat{r}_i \mathbf{x}_{i,j}\}_{i \in [d]}$ *can be found in the output representation* $\mathbf{H}^{(1)}$:

$$\mathbf{H}^{(1)} = \mathsf{TF}_1 \circ \mathbf{W}_E(\mathbf{E}) = \begin{pmatrix} \widetilde{\mathbf{x}}_1 & \widetilde{\mathbf{x}}_2 & \cdots & \widetilde{\mathbf{x}}_n & \widetilde{\mathbf{x}}_{n+1} & \cdots & \widetilde{\mathbf{x}}_{n+q} \\ y_1 & y_2 & \cdots & y_n & 0 & \cdots & 0 \\ \vdots & \vdots & \ddots & \vdots & \vdots \ddots & & \vdots \end{pmatrix}.$$

*Proof.* Here we first explain the key steps of our constructed transformer: the model first re-arrange the input entries with a input projection to divide the input data into $d$ subspace $\mathbf{W}_E$, each subspace includes an entry of $\mathbf{x}$ and the corresponding $y$ (step C.2), then use $h$ parameters $\{\mathbf{W}_{V_i}, \mathbf{W}_{K_i}, \mathbf{W}_{Q_i}\}_{i=1}^h$ to calculate $h$ queries, keys and values (step C.3), and compute the attention output for each head and concatenate them together (step C.4), finally use a projection matrix $\mathbf{W}_1$

rearrange the result, resulting the target output (step C.5):

$$\mathbf{E} = \begin{pmatrix} \mathbf{x}_1 & \mathbf{x}_2 & \cdots & \mathbf{x}_n & \mathbf{x}_{n+1} & \cdots & \mathbf{x}_{n+q} \\ y_1 & y_2 & \cdots & y_n & 0 & \cdots & 0 \end{pmatrix} \tag{C.1}$$

$$\xrightarrow[\mathbf{W}_E \in \mathbb{R}^{(d+1)\times d_{\mathsf{hid}}}]{\text{input projection}} \quad \mathbf{H} = \begin{pmatrix} \mathbf{x}_{1,1} & \mathbf{x}_{2,1} & \cdots & \mathbf{x}_{n,1} & \mathbf{x}_{(n+1),1} & \cdots & \mathbf{x}_{(n+q),1} \\ y_1 & y_2 & \cdots & y_n & 0 & \cdots & 0 \\ 0 & 0 & \cdots & 0 & 0 & \cdots & 0 \\ \vdots & \vdots & \ddots & \vdots & \vdots & \ddots & \vdots \end{pmatrix} \tag{C.2}$$

$$\xrightarrow[\mathbf{W}_{V_i},\mathbf{W}_{K_i},\mathbf{W}_{Q_i}\in\mathbb{R}^{3\times d_{\mathsf{hid}}}]{\text{compute } \mathbf{Q}_i, \mathbf{K}_i, \mathbf{V}_i} \quad \mathbf{K}_i = \frac{1}{n}\begin{pmatrix} 0 & \cdots & 0 & 0 & \cdots \\ 0 & \cdots & 0 & 0 & \cdots \\ y_1 & \cdots & y_n & 0 & \cdots \end{pmatrix}; \mathbf{Q}_i, \mathbf{V}_i = \begin{pmatrix} 0 & \cdots & 0 & 0 & \cdots \\ 0 & \cdots & 0 & 0 & \cdots \\ \mathbf{x}_{1,i} & \cdots & \mathbf{x}_{n,i} & \mathbf{x}_{(n+1),i} & \cdots \end{pmatrix} \tag{C.3}$$

$$\xrightarrow[\mathbf{H}+\text{Concat}\{\mathbf{V}_i\mathbf{M}\mathbf{K}_i^\top\mathbf{Q}_i\}]{\text{Attn}(\mathbf{W}_E(\mathbf{E}))} \quad \begin{pmatrix} \mathbf{x}_{1,1} & \mathbf{x}_{2,1} & \cdots & \mathbf{x}_{n,1} & \mathbf{x}_{(n+1),1} & \cdots & \mathbf{x}_{(n+q),1} \\ y_1 & y_2 & \cdots & y_n & 0 & 0 & 0 \\ \widetilde{\mathbf{x}}_{1,1} & \widetilde{\mathbf{x}}_{2,1} & \cdots & \widetilde{\mathbf{x}}_{n,1} & \widetilde{\mathbf{x}}_{(n+1),1} & \cdots & \widetilde{\mathbf{x}}_{(n+q),1} \\ \vdots & \vdots & \ddots & \vdots & \vdots & \ddots & \vdots \end{pmatrix} \tag{C.4}$$

$$\xrightarrow[\mathbf{W}_1\in\mathbb{R}^{d_{\mathsf{hid}}\times d_{\mathsf{hid}}}]{\mathbf{H}^{(1)}=\mathsf{TF}_1\circ\mathbf{W}_E(\mathbf{E})} \quad \begin{pmatrix} \widetilde{\mathbf{x}}_1 & \widetilde{\mathbf{x}}_2 & \cdots & \widetilde{\mathbf{x}}_n & \widetilde{\mathbf{x}}_{n+1} & \cdots & \widetilde{\mathbf{x}}_{n+q} \\ y_1 & y_2 & \cdots & y_n & 0 & \cdots & 0 \\ \vdots & \vdots & \ddots & \vdots & \vdots & \ddots & \vdots \end{pmatrix} \tag{C.5}$$

. The detailed parameters and calculation process for each step are as follows:

- we set $\mathbf{W}_E \in \mathbb{R}^{(d+1)\times d_{\mathsf{hid}}}$ to rearrange the entries:

$$\mathbf{W}_E = \begin{pmatrix} \mathbb{1}[1] & \mathbb{1}[d+1] & \mathbf{0} & \mathbb{1}[2] & \mathbb{1}[d+1] & \mathbf{0} & \cdots & \mathbb{1}[d] & \mathbb{1}[d+1] & \mathbf{0} \end{pmatrix}^\top,$$

where $\mathbb{1}[k]$ is an $1 \times d_{\mathsf{hid}}$ vector with 1 at $i$-th entry and 0 elsewhere, such that

$$\mathbf{H} = \mathbf{W}_E\mathbf{E} = \begin{pmatrix} \mathbf{x}_{1,1} & \mathbf{x}_{2,1} & \cdots & \mathbf{x}_{n,1} & \mathbf{x}_{(n+1),1} & \cdots & \mathbf{x}_{(n+q),1} \\ y_1 & y_2 & \cdots & y_n & 0 & \cdots & 0 \\ 0 & 0 & \cdots & 0 & 0 & \cdots & 0 \\ \mathbf{x}_{1,2} & \mathbf{x}_{2,2} & \cdots & \mathbf{x}_{n,2} & \mathbf{x}_{(n+1),2} & \cdots & \mathbf{x}_{(n+q),2} \\ \vdots & \vdots & \ddots & \vdots & \vdots & \ddots & \vdots \end{pmatrix}.$$

- we set $\mathbf{W}_{V_i}, \mathbf{W}_{K_i}, \mathbf{W}_{Q_i} \in \mathbb{R}^{3\times d_{\mathsf{hid}}}$ for values, keys and queries:

$$\mathbf{W}_{K_i} = \frac{1}{n}\begin{pmatrix} \mathbf{0} \\ \mathbf{0} \\ \mathbb{1}[3i-1] \end{pmatrix}; \quad \mathbf{W}_{V_i}, \mathbf{W}_{Q_i} = \begin{pmatrix} \mathbf{0} \\ \mathbf{0} \\ \mathbb{1}[3i-2] \end{pmatrix},$$

such that the $i$-th head extract $i$-th entry of $\mathbf{x}$ and corresponding $y$

$$\mathbf{K}_i = \frac{1}{n}\begin{pmatrix} \mathbf{0} \\ \mathbf{0} \\ \mathbb{1}[3i-1] \end{pmatrix} \begin{pmatrix} \mathbf{x}_{1,1} & \mathbf{x}_{2,1} & \cdots & \mathbf{x}_{n,1} & \mathbf{x}_{(n+1),1} & \cdots \\ y_1 & y_2 & \cdots & y_n & 0 & \cdots \\ 0 & 0 & \cdots & 0 & 0 & \cdots \\ \mathbf{x}_{1,2} & \mathbf{x}_{2,2} & \cdots & \mathbf{x}_{n,2} & \mathbf{x}_{(n+1),2} & \cdots \\ \vdots & \vdots & \ddots & \vdots & \vdots & \ddots \end{pmatrix} = \frac{1}{n}\begin{pmatrix} 0 & \cdots & 0 & 0 & \cdots \\ 0 & \cdots & 0 & 0 & \cdots \\ y_1 & \cdots & y_n & 0 & \cdots \end{pmatrix},$$

$$\mathbf{Q}_i, \mathbf{V}_i = \begin{pmatrix} \mathbf{0} \\ \mathbf{0} \\ \mathbb{1}[3i-2] \end{pmatrix} \begin{pmatrix} \mathbf{x}_{1,1} & \mathbf{x}_{2,1} & \cdots & \mathbf{x}_{n,1} & \mathbf{x}_{(n+1),1} & \cdots \\ y_1 & y_2 & \cdots & y_n & 0 & \cdots \\ 0 & 0 & \cdots & 0 & 0 & \cdots \\ \mathbf{x}_{1,2} & \mathbf{x}_{2,2} & \cdots & \mathbf{x}_{n,2} & \mathbf{x}_{(n+1),2} & \cdots \\ \vdots & \vdots & \ddots & \vdots & \vdots & \ddots \end{pmatrix} = \begin{pmatrix} 0 & \cdots & 0 & 0 & \cdots \\ 0 & \cdots & 0 & 0 & \cdots \\ \mathbf{x}_{1,i} & \cdots & \mathbf{x}_{n,i} & \mathbf{x}_{(n+1),i} & \cdots \end{pmatrix},$$

$$\mathbf{V}_i\mathbf{M}\mathbf{K}_i^\top\mathbf{Q}_i = \begin{pmatrix} 0 & \cdots & 0 & 0 & \cdots & 0 \\ 0 & \cdots & 0 & 0 & \cdots & 0 \\ \mathbf{x}_{1,i} & \cdots & \mathbf{x}_{n,i} & \mathbf{x}_{(n+1),i} & \cdots & \mathbf{x}_{(n+q),i} \end{pmatrix} \begin{pmatrix} \mathbf{I}_n & \mathbf{0} \\ \mathbf{0} & \mathbf{0} \end{pmatrix}.$$

$$\begin{pmatrix} 0 & \cdots & 0 & 0 & \cdots & 0 \\ 0 & \cdots & 0 & 0 & \cdots & 0 \\ y_1 & \cdots & y_n & 0 & \cdots & 0 \end{pmatrix}^\top \begin{pmatrix} 0 & \cdots & 0 & 0 & \cdots & 0 \\ 0 & \cdots & 0 & 0 & \cdots & 0 \\ \mathbf{x}_{1,i} & \cdots & \mathbf{x}_{n,i} & \mathbf{x}_{(n+1),i} & \cdots & \mathbf{x}_{(n+q),i} \end{pmatrix}$$

$$= \begin{pmatrix} 0 & \cdots & 0 & 0 & \cdots & 0 \\ 0 & \cdots & 0 & 0 & \cdots & 0 \\ \widetilde{\mathbf{x}}_{1,i} & \cdots & \widetilde{\mathbf{x}}_{n,i} & \widetilde{\mathbf{x}}_{(n+1),i} & \cdots & \widetilde{\mathbf{x}}_{(n+q),i} \end{pmatrix}.$$

- Then concatenate the output of each head $\{\mathbf{V}_i\mathbf{M}\mathbf{K}_i^\top\mathbf{Q}_i\}_{i=1}^h$ together with residue:

$$\mathbf{H} + \mathsf{Concat}[\{\mathbf{V}_i\mathbf{M}\mathbf{K}_i^\top\mathbf{Q}_i\}_{i=1}^h] = \begin{pmatrix} \mathbf{x}_{1,1} & \mathbf{x}_{2,1} & \cdots & \mathbf{x}_{n,1} & \mathbf{x}_{(n+1),1} & \cdots & \mathbf{x}_{(n+q),1} \\ y_1 & y_2 & \cdots & y_n & 0 & \cdots & 0 \\ \widetilde{\mathbf{x}}_{1,1} & \widetilde{\mathbf{x}}_{2,1} & \cdots & \widetilde{\mathbf{x}}_{n,1} & \widetilde{\mathbf{x}}_{(n+1),1} & \cdots & \widetilde{\mathbf{x}}_{(n+q),1} \\ \vdots & \vdots & \ddots & \vdots & \vdots & \ddots & \vdots \end{pmatrix}.$$
(C.6)

- Finally, $\mathbf{W}_1$ is applied to rearrange the entries:

$$\mathbf{W}_1 = \begin{pmatrix} \mathbb{1}[3] & \cdots & \mathbb{1}[3d] & \mathbb{1}[2] & \cdots \end{pmatrix}^\top,$$

where the first $\cdots$ implies the omitted $d-2$ vectors $\{\mathbb{1}[3i]|i = 2,3,\ldots,(d-1)\}$, the second $\cdots$ implies arbitrary values, then resulting the final output:

$$\mathbf{H}^{(1)} = \mathbf{W}_1\big[\mathbf{H} + \mathsf{Concat}[\{\mathbf{V}_i\mathbf{M}\mathbf{K}_i^\top\mathbf{Q}_i\}_{i=1}^h]\big] = \begin{pmatrix} \widetilde{\mathbf{x}}_1 & \widetilde{\mathbf{x}}_2 & \cdots & \widetilde{\mathbf{x}}_n & \widetilde{\mathbf{x}}_{n+1} & \cdots & \widetilde{\mathbf{x}}_{n+q} \\ y_1 & y_2 & \cdots & y_n & 0 & \cdots & 0 \\ \vdots & \vdots & \ddots & \vdots & \vdots & \ddots & \vdots \end{pmatrix}.$$

in this way we construct a transformer that can apply Alg. 1 so that each of the enhanced data $\{\widehat{r}_i\mathbf{x}_{i,j}\}_{i\in[d]}$ can be found in the output representation $\mathbf{H}^{(1)}$. $\qquad\square$

## C.2 Proof for Proposition 4.2

**Proposition C.2** (Restate of Proposition 4.2). *There exists a transformer with $k$ layers, $1$ head, $d_{\mathsf{hid}} = 3d$, let $\{(\widetilde{\mathbf{x}}_i, \widehat{y}_{(i)}^\ell)\}_{i=1}^{n+1}$ be the $\ell$-th layer input data entry, then it holds that $\widehat{y}_{(n+1)}^\ell = \langle \mathbf{w}_{\mathsf{gd}}^\ell, \widetilde{\mathbf{x}}_{n+1}\rangle$, where $\mathbf{w}_{\mathsf{gd}}$ is defined as $\mathbf{w}_{\mathsf{gd}}^0 = 0$ and as follows for $\ell = 0,\ldots,k-1$:*

$$\mathbf{w}_{\mathsf{gd}}^{\ell+1} = \mathbf{w}_{\mathsf{gd}}^\ell - \eta\nabla\widetilde{L}(\mathbf{w}_{\mathsf{gd}}^\ell), \quad where \quad \widetilde{L}(\mathbf{w}) = \frac{1}{2n}\sum_{i=1}^n(y_i - \langle\mathbf{w}, \widetilde{\mathbf{x}}_i\rangle)^2.$$

*Proof.* Here we directly provide the parameters $\mathbf{W}_V^\ell, \mathbf{W}_K^\ell, \mathbf{W}_Q^\ell \in \mathbb{R}^{d_{\mathsf{hid}}\times d_{\mathsf{hid}}}$ and $\mathbf{W}_1^\ell \in \mathbb{R}^{d_{\mathsf{hid}}\times d_{\mathsf{hid}}}$ for each layer $\mathsf{TF}_\ell$,

$$\mathbf{W}_V^\ell = -\frac{\eta}{n}\begin{pmatrix} \mathbf{0} & \mathbf{0} \\ \mathbf{0} & 1 \end{pmatrix}; \quad \mathbf{W}_K^\ell, \mathbf{W}_Q^\ell = \begin{pmatrix} \mathbf{I}_{d\times d} & \mathbf{0} \\ \mathbf{0} & \mathbf{0} \end{pmatrix}; \quad \mathbf{W}_1^\ell = \mathbf{I}_{d_{\mathsf{hid}}\times d_{\mathsf{hid}}} \qquad (\text{C.7})$$

As we set $\mathbf{W}_1^\ell$ as the identity matrix, we can ignore it and then apply Lemma 1 in [3]. By replacing $(\mathbf{W}_K^{\ell\top}\mathbf{W}_Q^\ell)$ as $\mathbf{Q}_i$ and $\mathbf{W}_V^\ell$ with $\mathbf{P}_i$, then it holds that $\widehat{y}_{(n+1)}^\ell = \langle\mathbf{w}_{\mathsf{gd}}^\ell, \widetilde{\mathbf{x}}_{n+1}\rangle$, where $\mathbf{w}_{\mathsf{gd}}$ is defined as $\mathbf{w}_{\mathsf{gd}}^0 = 0$ and as follows for $\ell = 0,\ldots,k-1$:

$$\mathbf{w}_{\mathsf{gd}}^{\ell+1} = \mathbf{w}_{\mathsf{gd}}^\ell - \eta\nabla\widetilde{L}(\mathbf{w}_{\mathsf{gd}}^\ell), \quad where \quad \widetilde{L}(\mathbf{w}) = \frac{1}{2n}\sum_{i=1}^n(y_i - \langle\mathbf{w}, \widetilde{\mathbf{x}}_i\rangle)^2.$$

$\qquad\square$

# D  Proof of Theorem 5.1

To simplify the notations, we use $\widehat{\mathbf{w}}_t$ to denote $\widetilde{\mathbf{w}}_{\mathrm{gd}}^t$. We first prove that with a high probability, there exists a $\overline{\mathbf{R}} \in \mathbb{R}^{d \times d}$ such that $\overline{\mathbf{R}}\widehat{\mathbf{R}} = \widehat{\mathbf{R}}\overline{\mathbf{R}} = \mathbf{I}_s$, where $\mathbf{I}_s = \mathrm{diag}\{a_1, \ldots, a_d\}$ with $a_j = 1_{\{j \in \mathcal{S}\}}$.

**Lemma D.1.** *Denote* $\mathbf{R} = \mathrm{diag}\{r_1, \ldots, r_d\}$, *where* $r_j = \sum_{i=1}^d w_i^\star \Sigma_{ij}$. *Suppose* $n \geq \mathcal{O}(\log{(d/\delta)})$, *then for any* $\delta \in (0, 1)$ *with probability at least* $1 - \delta$, *we have*

$$\|\widehat{\mathbf{R}} - \mathbf{R}\|_2 \lesssim K \cdot \sqrt{\frac{s \log{(d/\delta)}}{n}},$$

*where* $K := C\big(\max_i \Sigma_{ii} + \sigma^2\big)$, *where* $C$ *is an absolute constant.*

**Lemma D.2.** *Define the event* $\mathcal{E}_R$ *by* $\mathcal{E}_R = \big\{|\widehat{r}|_i \geq \frac{1}{2}|r_i|, \ \forall i \in \mathcal{S}\big\}$. *Suppose that* $n \gtrsim s \log{(d/\delta)}/\beta^2$, *then* $\mathbb{P}(\mathcal{E}_1) \geq 1 - \delta$.

We define $\overline{\mathbf{R}}$ by $\overline{\mathbf{R}} = \mathrm{diag}\{\overline{r}_1, \ldots, \overline{r}_d\}$, where $\overline{r}_j$ is given by

$$\overline{r}_j = \begin{cases} 0 & j \notin \mathcal{S}, \\ 1/\widehat{r}_j & j \in \mathcal{S}. \end{cases}$$

It is easy to see $\overline{\mathbf{R}}\widehat{\mathbf{R}} = \widehat{\mathbf{R}}\overline{\mathbf{R}} = \mathbf{I}_s$. On the event $\mathcal{E}_1$, we have that $\|\overline{\mathbf{R}}\| \lesssim 1/\beta$. Hereafter, we condition on $\mathcal{E}_1$.

## D.1  Bias-variance Decomposition

Let $\widetilde{\mathbf{X}} = \mathbf{X}\widehat{\mathbf{R}}$ with $\widetilde{\mathbf{x}}_i = \widehat{\mathbf{R}}\mathbf{x}_i$. For $\widehat{\mathbf{w}}_t$, we have

$$\begin{aligned}
\widehat{\mathbf{w}}_{t+1} - \overline{\mathbf{R}}\mathbf{w}^\star &= \widehat{\mathbf{w}}_t - \overline{\mathbf{R}}\mathbf{w}^\star - \eta \cdot \frac{1}{n}\sum_{i=1}^n \widetilde{\mathbf{x}}_i\big(\widetilde{\mathbf{x}}_i^\top \widehat{\mathbf{w}}_t - y_i\big) \\
&= \widehat{\mathbf{w}}_t - \overline{\mathbf{R}}\mathbf{w}^\star - \eta \cdot \frac{1}{n}\sum_{i=1}^n \widetilde{\mathbf{x}}_i\big(\widetilde{\mathbf{x}}_i^\top \widehat{\mathbf{w}}_t - \widetilde{\mathbf{x}}_i^\top \overline{\mathbf{R}}\mathbf{w}^\star + \epsilon\big) \\
&= \big(\mathbf{I} - \eta\widehat{\mathbf{\Sigma}}\big)\big(\widehat{\mathbf{w}}_t - \overline{\mathbf{R}}\mathbf{w}^\star\big) + \eta \cdot \frac{1}{n}\widetilde{\mathbf{X}}^\top \epsilon.
\end{aligned}$$

Hence, we have

$$\widehat{\mathbf{w}}_t = \Big(\mathbf{I} - \big(\mathbf{I} - \eta\widehat{\mathbf{\Sigma}}\big)^t\Big)\overline{\mathbf{R}}\mathbf{w}^\star + \frac{1}{n}\sum_{i=1}^t \big(\mathbf{I} - \eta\widehat{\mathbf{\Sigma}}\big)^{i-1}\widetilde{\mathbf{X}}^\top \epsilon. \tag{D.1}$$

We can decompose the risk $L(\widehat{\mathbf{w}}_t)$ by

$$\begin{aligned}
\mathcal{E}(\widehat{\mathbf{w}}_t) &= \mathbb{E}_{(\mathbf{x},y)\sim\mathsf{P}}\Big[\big(\langle\widehat{\mathbf{R}}\mathbf{x}, \widehat{\mathbf{w}}_t\rangle - \langle\widehat{\mathbf{R}}\mathbf{x}, \overline{\mathbf{R}}\mathbf{w}^\star\rangle - \epsilon\big)^2\Big] - \sigma^2 \tag{D.2} \\
&= \mathbb{E}_{(\mathbf{x},y)\sim\mathsf{P}}\Big[\big(\langle\widehat{\mathbf{R}}\mathbf{x}, \widehat{\mathbf{w}}_t\rangle - \langle\widehat{\mathbf{R}}\mathbf{x}, \overline{\mathbf{R}}\mathbf{w}^\star\rangle\big)^2\Big] \\
&= \Big\|\mathbf{\Sigma}^{1/2}\widehat{\mathbf{R}}\big(\widehat{\mathbf{w}}_t - \overline{\mathbf{R}}\mathbf{w}^\star\big)\Big\|_2^2 \\
&= \Big\|\mathbf{\Sigma}^{1/2}\widehat{\mathbf{R}}\Big(-\big(\mathbf{I} - \eta\widehat{\mathbf{\Sigma}}\big)^t\overline{\mathbf{R}}\mathbf{w}^\star + \eta \cdot \frac{1}{n}\sum_{i=1}^t \big(\mathbf{I} - \eta\widehat{\mathbf{\Sigma}}\big)^{i-1}\widetilde{\mathbf{X}}^\top \epsilon\Big)\Big\|_2^2 \\
&= \underbrace{\Big\|\mathbf{\Sigma}^{1/2}\widehat{\mathbf{R}}\big(\mathbf{I} - \eta\widehat{\mathbf{\Sigma}}\big)^t\overline{\mathbf{R}}\mathbf{w}^\star\Big\|_2^2}_{\text{Bias}} + \eta^2\underbrace{\Big\|\mathbf{\Sigma}^{1/2}\widehat{\mathbf{R}}\Big(\frac{1}{n}\sum_{i=1}^t \big(\mathbf{I} - \eta\widehat{\mathbf{\Sigma}}\big)^{i-1}\widetilde{\mathbf{X}}^\top \epsilon\Big)\Big\|_2^2}_{\text{Variance}}. \tag{D.3}
\end{aligned}$$

Next, we present some lemmas.

**Lemma D.3** (Theorem 9 in Bartlett et al. [9])**.** *There is an absolute constant $c$ such that for any $\delta \in (0,1)$ with probability at least $1 - \delta$,*

$$\|\widehat{\boldsymbol{\Sigma}} - \boldsymbol{\Sigma}\|_2 \leq c\|\boldsymbol{\Sigma}\|_2 \cdot \max\left\{\sqrt{\frac{r(\boldsymbol{\Sigma})}{n}}, \frac{r(\boldsymbol{\Sigma})}{n}, \sqrt{\frac{\log\left(1/\delta\right)}{n}}, \frac{\log\left(1/\delta\right)}{n}\right\},$$

*where $r(\boldsymbol{\Sigma}) = \mathrm{Tr}(\boldsymbol{\Sigma})/\lambda_1$.*

**Lemma D.4.** *With probability at least $1 - \delta$, we have*

$$\|\widehat{\mathbf{R}}\widehat{\boldsymbol{\Sigma}}\widehat{\mathbf{R}} - \mathbf{R}\boldsymbol{\Sigma}\mathbf{R}\|_2 \lesssim \sqrt{s} \cdot \mathrm{poly}(\log\left(d/\delta\right)) \cdot \left(\sqrt{\frac{r(\mathbf{R}\boldsymbol{\Sigma}\mathbf{R})}{n}} + \frac{\sqrt{r(\boldsymbol{\Sigma})} + r(\mathbf{R}\boldsymbol{\Sigma}\mathbf{R})}{n} + \frac{r(\boldsymbol{\Sigma})}{n^{3/2}}\right).$$

*As a result, when $n \gtrsim st^2\left(r^{2/3}(\boldsymbol{\Sigma}) + r(\mathbf{R}\boldsymbol{\Sigma}\mathbf{R})\right) \cdot \mathrm{poly}(\log\left(d/\delta\right))$, with probability at least $1 - \delta$, we have*

$$\|\widehat{\mathbf{R}}\widehat{\boldsymbol{\Sigma}}\widehat{\mathbf{R}} - \mathbf{R}\boldsymbol{\Sigma}\mathbf{R}\|_2 \leq 1/t.$$

We define the event $\mathcal{E}_2$ as follows:

$$\mathcal{E}_2 := \left\{\|\mathbf{R}\boldsymbol{\Sigma}\mathbf{R}\|_2 \lesssim \widetilde{\alpha}(n,\delta) \leq 1/t\right\},$$

where

$$\widetilde{\alpha}(n,\delta) = \sqrt{s} \cdot \mathrm{poly}(\log\left(d/\delta\right)) \cdot \left(\sqrt{\frac{r(\mathbf{R}\boldsymbol{\Sigma}\mathbf{R})}{n}} + \frac{\sqrt{r(\boldsymbol{\Sigma})} + r(\mathbf{R}\boldsymbol{\Sigma}\mathbf{R})}{n} + \frac{r(\boldsymbol{\Sigma})}{n^{3/2}}\right).$$

By Lemma D.4, $\mathbb{P}(\mathcal{E}_2) \geq 1 - \delta$. Hereafter, we condition on $\mathcal{E}_1 \cap \mathcal{E}_2$.

## D.2 Bounding the Bias

On $\mathcal{E}_1 \cap \mathcal{E}_2$, we have

$$\begin{aligned}
\mathrm{Bias} &= \left\|\boldsymbol{\Sigma}^{1/2}\widehat{\mathbf{R}}\left(\mathbf{I} - \eta\widehat{\boldsymbol{\Sigma}}\right)^t \overline{\mathbf{R}}\mathbf{w}^\star\right\|_2^2 \\
&= \mathbf{w}^{\star\top}\overline{\mathbf{R}}\left(\mathbf{I} - \eta\widehat{\boldsymbol{\Sigma}}\right)^t \widehat{\mathbf{R}}\boldsymbol{\Sigma}\widehat{\mathbf{R}}\left(\mathbf{I} - \eta\widehat{\boldsymbol{\Sigma}}\right)^t \overline{\mathbf{R}}\mathbf{w}^\star \\
&= \underbrace{\mathbf{w}^{\star\top}\overline{\mathbf{R}}\left(\mathbf{I} - \eta\widehat{\boldsymbol{\Sigma}}\right)^t \widehat{\mathbf{R}}\left(\boldsymbol{\Sigma} - \widehat{\boldsymbol{\Sigma}}\right)\widehat{\mathbf{R}}\left(\mathbf{I} - \eta\widehat{\boldsymbol{\Sigma}}\right)^t \overline{\mathbf{R}}\mathbf{w}^\star}_{\mathrm{I}} + \underbrace{\mathbf{w}^{\star\top}\overline{\mathbf{R}}\left(\mathbf{I} - \eta\widehat{\boldsymbol{\Sigma}}\right)^t \widehat{\mathbf{R}}\widehat{\boldsymbol{\Sigma}}\widehat{\mathbf{R}}\left(\mathbf{I} - \eta\widehat{\boldsymbol{\Sigma}}\right)^t \overline{\mathbf{R}}\mathbf{w}^\star}_{\mathrm{II}}..
\end{aligned}$$

$$\text{(D.4)}$$

**Lemma D.5.** *On $\mathcal{E}_1 \cap \mathcal{E}_2$, we have*

$$\mathrm{I} \lesssim \frac{1}{t\beta^2}$$

*and*

$$\mathrm{II} \lesssim \frac{1}{\eta t\beta^2}.$$

*hold with probability at least $1 - \delta$.*

By Lemma D.5, we obtain that with probability at least $1 - \delta$,

$$\mathrm{Bias} \lesssim \mathrm{I} + \mathrm{II} \leq \frac{1}{t\beta^2} + \frac{1}{\eta t\beta^2} \lesssim \frac{1}{\eta t\beta^2} \tag{D.5}$$

where the last inequality is by $\eta \lesssim 1/\|\boldsymbol{\Sigma}\| \lesssim 1$.

## D.3 Bounding the Variance

$$
\begin{aligned}
\text{Variance} &= \eta^2 \left\| \mathbf{\Sigma}^{1/2} \widehat{\mathbf{R}} \left( \frac{1}{n} \sum_{i=1}^{t} \left( \mathbf{I} - \eta \widehat{\mathbf{\Sigma}} \right)^{i-1} \widetilde{\mathbf{X}}^\top \epsilon \right) \right\|_2^2 \\
&= \frac{\eta^2}{n^2} \epsilon^\top \mathbf{X} \widehat{\mathbf{R}} \sum_{i=1}^{t} \left( \mathbf{I} - \eta \widehat{\mathbf{\Sigma}} \right)^{i-1} \widehat{\mathbf{R}} \mathbf{\Sigma} \widehat{\mathbf{R}} \sum_{i=1}^{t} \left( \mathbf{I} - \eta \widehat{\mathbf{\Sigma}} \right)^{i-1} \widehat{\mathbf{R}} \mathbf{X}^\top \epsilon \\
&= \underbrace{\frac{\eta^2}{n^2} \epsilon^\top \mathbf{X} \widehat{\mathbf{R}} \sum_{i=1}^{t} \left( \mathbf{I} - \eta \widehat{\mathbf{\Sigma}} \right)^{i-1} \widehat{\mathbf{R}} \left( \mathbf{\Sigma} - \widehat{\mathbf{\Sigma}} \right) \widehat{\mathbf{R}} \sum_{i=1}^{t} \left( \mathbf{I} - \eta \widehat{\mathbf{\Sigma}} \right)^{i-1} \widehat{\mathbf{R}} \mathbf{X}^\top \epsilon}_{\text{I}} \\
&+ \underbrace{\frac{\eta^2}{n^2} \epsilon^\top \mathbf{X} \widehat{\mathbf{R}} \sum_{i=1}^{t} \left( \mathbf{I} - \eta \widehat{\mathbf{\Sigma}} \right)^{i-1} \widehat{\mathbf{R}} \widehat{\mathbf{\Sigma}} \widehat{\mathbf{R}} \sum_{i=1}^{t} \left( \mathbf{I} - \eta \widehat{\mathbf{\Sigma}} \right)^{i-1} \widehat{\mathbf{R}} \mathbf{X}^\top \epsilon}_{\text{II}} .
\end{aligned} \tag{D.6}
$$

**Lemma D.6.** *On $\mathcal{E}_1 \cap \mathcal{E}_2$, with probability at least $1 - \delta$, we have*

$$
\text{I} \lesssim \frac{\eta^2 t}{n^2} \cdot \left\| \widehat{\mathbf{R}} \mathbf{X}^\top \epsilon \right\|_2^2
$$

*and*

$$
\text{II} \lesssim \frac{\eta t \log t}{n^2} \cdot \left\| \widehat{\mathbf{R}} \mathbf{X}^\top \epsilon \right\|_2^2 .
$$

By applying Lemma D.6 to Eq.(D.6), we obtain that

$$
\text{Variance} = \text{I} + \text{II} \lesssim \frac{\eta^2 t}{n^2} \cdot \left\| \widehat{\mathbf{R}} \mathbf{X}^\top \epsilon \right\|_2^2 + \frac{\eta t \log t}{n^2} \cdot \left\| \widehat{\mathbf{R}} \mathbf{X}^\top \epsilon \right\|_2^2 \lesssim \frac{\eta t \log t}{n^2} \cdot \left\| \widehat{\mathbf{R}} \mathbf{X}^\top \epsilon \right\|_2^2 . \tag{D.7}
$$

**Lemma D.7.** *with probability at least $1 - \delta$, we have*

$$
\left\| \frac{1}{n} \cdot \widehat{\mathbf{R}} \mathbf{X}^\top \epsilon \right\|_2^2 \lesssim \frac{\sigma^2 \text{Tr}(\mathbf{R} \mathbf{\Sigma} \mathbf{R}) \log (d/\delta)}{n} + \frac{\sigma^2 s \text{Tr}(\mathbf{\Sigma}) \log^2 (d/\delta)}{n^2}
$$

By applying Lemma D.7 to Eq.(D.7), we obtain that

$$
\text{Variance} \lesssim \eta t \log t \cdot \left( \frac{\sigma^2 \text{Tr}(\mathbf{R} \mathbf{\Sigma} \mathbf{R}) \log (d/\delta)}{n} + \frac{\sigma^2 s \text{Tr}(\mathbf{\Sigma}) \log^2 (d/\delta)}{n^2} \right) . \tag{D.8}
$$

## D.4 Final Bound

Combining Eq.(D.5) and Eq.(D.8), we obtain that

$$
\begin{aligned}
\mathcal{E}(\widehat{\mathbf{w}}_t) &\leq \frac{1}{\eta t \beta^2} + \eta t \log t \cdot \left( \frac{\sigma^2 \text{Tr}(\mathbf{R} \mathbf{\Sigma} \mathbf{R}) \log (d/\delta)}{n} + \frac{\sigma^2 s \text{Tr}(\mathbf{\Sigma}) \log^2 (d/\delta)}{n^2} \right) \\
&\lesssim \frac{\log t}{\beta} \sqrt{\frac{\sigma^2 \text{Tr}(\mathbf{R} \mathbf{\Sigma} \mathbf{R}) \log (d/\delta)}{n} + \frac{\sigma^2 s \text{Tr}(\mathbf{\Sigma}) \log^2 (d/\delta)}{n^2}},
\end{aligned}
$$

when $\eta t \simeq \frac{1}{\beta} \cdot \left( \frac{\sigma^2 \text{Tr}(\mathbf{R} \mathbf{\Sigma} \mathbf{R}) \log (d/\delta)}{n} + \frac{\sigma^2 s \text{Tr}(\mathbf{\Sigma}) \log^2 (d/\delta)}{n^2} \right)^{-1/2}$.

## D.5 Proof for Appendix D

*Proof of Lemma D.1.* Since $y_i = \sum_{j=1}^{d} w_j^\star x_{ij} + \epsilon_i$, then we have

$$
\widehat{r}_i = \frac{1}{n} \sum_{j=1}^{n} x_{ji} y_j = \frac{1}{n} \sum_{j=1}^{n} x_{ji} \cdot \left( \sum_{k=1}^{d} w_k^\star x_{jk} + \epsilon_j \right) = \sum_{k=1}^{d} \frac{w_k^\star}{n} \sum_{j=1}^{n} x_{jk} x_{ji} + \frac{1}{n} \sum_{j=1}^{n} x_{ji} \epsilon_j . \tag{D.9}
$$

Since $x_{ji} \sim \mathsf{N}(0, \Sigma_{ii})$ for any $i, j$, by Lemma 2.7.7 in Vershynin [46], there exists an absolute constant $C$ such that $x_{jk}x_{ji}$ is a sub-exponential random variable with

$$\|x_{jk}x_{ji}\|_{\Psi_1} \leq C\sqrt{\Sigma_{kk}\Sigma_{ii}} \leq K,$$

where $\|\cdot\|_{\Psi_1}$ denotes the sub-exponential norm and the last inequality comes from the definition of $K$. By applying Bernstein's inequality [46, Theorem 2.8.1], we have

$$\left|\frac{1}{n}\sum_{j=1}^{n}x_{jk}x_{ji} - \mathbb{E}[x_{1k}x_{1i}]\right| = \left|\frac{1}{n}\sum_{j=1}^{n}x_{jk}x_{ji} - \Sigma_{ki}\right|$$

$$\leq K \cdot \max\left\{\sqrt{\frac{\log(d/\delta)}{n}}, \frac{\log(d/\delta)}{n}\right\}$$

$$= K \cdot \sqrt{\frac{\log(d/\delta)}{n}}, \tag{D.10}$$

where the last equality due to $n \geq \mathcal{O}(\log(d/\delta))$. We also note that $x_{ji}\epsilon_j$ is a sub-exponential random variable with $\|x_{ji}\epsilon_j\|_{\Psi_1} \leq K$. Hence, we also have

$$\left|\frac{1}{n}\sum_{j=1}x_{ji}\epsilon_j\right| \lesssim K \cdot \sqrt{\frac{\log(d/\delta)}{n}}. \tag{D.11}$$

Combining Eq.(D.9), Eq.(D.10) and Eq.(D.11), we have

$$|\widehat{r}_i - r_i| \lesssim K \cdot \sqrt{\frac{\log(d/\delta)}{n}}\sum_{k=1}^{d}|w_k^\star| + K \cdot \sqrt{\frac{\log(d/\delta)}{n}} = (\|w^\star\|_1 + 1)K \cdot \sqrt{\frac{\log(d/\delta)}{n}}.$$

By definition of $\widehat{\mathbf{R}}$ and $\mathbf{R}$, we obtain

$$\|\widehat{\mathbf{R}} - \mathbf{R}\|_2 = \max_i|\widehat{r}_i - r_i| \leq K(\|w^\star\|_1 + 1) \cdot \sqrt{\frac{\log(d/\delta)}{n}}$$

$$\leq K\left(\sqrt{s\|\mathbf{w}^\star\|_2^2} + 1\right) \cdot \sqrt{\frac{\log(d/\delta)}{n}} \lesssim K \cdot \sqrt{\frac{s\log(d/\delta)}{n}},$$

which completes the proof. $\qquad\square$

*Proof of Lemma D.2.* By Lemma D.1, for any $j \in \mathcal{S}$, with probability at least $1 - \delta$, we have

$$|r_i - \widehat{r}_j| \lesssim \sqrt{\frac{s\log(d/\delta)}{n}} \lesssim \beta/2 \leq |r_j|/2, \tag{D.12}$$

where the last inequality is due to the definition of $\beta$. $\qquad\square$

*Proof of Lemma D.4.* We can decompose $\|\widehat{\mathbf{R}}\widehat{\mathbf{\Sigma}}\widehat{\mathbf{R}} - \mathbf{R}\mathbf{\Sigma}\mathbf{R}\|_2$ as follows:

$$\|\widehat{\mathbf{R}}\widehat{\mathbf{\Sigma}}\widehat{\mathbf{R}} - \mathbf{R}\mathbf{\Sigma}\mathbf{R}\|_2 = \|\widehat{\mathbf{R}}\widehat{\mathbf{\Sigma}}\widehat{\mathbf{R}} - \mathbf{R}\widehat{\mathbf{\Sigma}}\widehat{\mathbf{R}} + \mathbf{R}\widehat{\mathbf{\Sigma}}\widehat{\mathbf{R}} - \mathbf{R}\mathbf{\Sigma}\widehat{\mathbf{R}} + \mathbf{R}\mathbf{\Sigma}\widehat{\mathbf{R}} - \mathbf{R}\mathbf{\Sigma}\mathbf{R}\|_2$$

$$\leq \underbrace{\|\widehat{\mathbf{R}}\widehat{\mathbf{\Sigma}}\widehat{\mathbf{R}} - \mathbf{R}\widehat{\mathbf{\Sigma}}\widehat{\mathbf{R}}\|_2}_{\text{I}} + \underbrace{\|\mathbf{R}\widehat{\mathbf{\Sigma}}\widehat{\mathbf{R}} - \mathbf{R}\mathbf{\Sigma}\widehat{\mathbf{R}}\|_2}_{\text{II}} + \underbrace{\|\mathbf{R}\mathbf{\Sigma}\widehat{\mathbf{R}} - \mathbf{R}\mathbf{\Sigma}\mathbf{R}\|_2}_{\text{III}}. \tag{D.13}$$

Next, we proof the bound for I, II and III separately.

For term I,

$$\text{I} = \|\widehat{\mathbf{R}}\widehat{\mathbf{\Sigma}}\widehat{\mathbf{R}} - \mathbf{R}\widehat{\mathbf{\Sigma}}\widehat{\mathbf{R}}\|_2 = \|\left(\widehat{\mathbf{R}} - \mathbf{R}\right)\widehat{\mathbf{\Sigma}}\widehat{\mathbf{R}}\|_2$$

$$\leq \|\widehat{\mathbf{R}} - \mathbf{R}\|_2 \cdot \|\widehat{\mathbf{\Sigma}}\|_2 \cdot \|\widehat{\mathbf{R}}\|_2$$

$$\leq \|\widehat{\mathbf{R}} - \mathbf{R}\|_2 \cdot \left(\|\mathbf{\Sigma}\|_2 + \|\widehat{\mathbf{\Sigma}} - \mathbf{\Sigma}\|_2\right) \cdot \left(\|\mathbf{R}\|_2 + \|\mathbf{R} - \widehat{\mathbf{R}}\|_2\right), \tag{D.14}$$

where the last line is due to triangle inequality. By Lemma D.3, with probability at least $1 - \delta/3$, we have

$$\|\widehat{\boldsymbol{\Sigma}} - \boldsymbol{\Sigma}\|_2 \lesssim \|\boldsymbol{\Sigma}\|_2 \cdot \max\left\{ \sqrt{\frac{r(\boldsymbol{\Sigma})}{n}}, \frac{r(\boldsymbol{\Sigma})}{n}, \sqrt{\frac{\log(1/\delta)}{n}}, \frac{\log(1/\delta)}{n} \right\}$$

$$\lesssim \|\boldsymbol{\Sigma}\|_2 \cdot \max\left\{ \sqrt{\frac{r(\boldsymbol{\Sigma}) + \log(1/\delta)}{n}}, \frac{r(\boldsymbol{\Sigma}) + \log(1/\delta)}{n} \right\}. \qquad \text{(D.15)}$$

By Lemma D.1, we obtain that

$$\|\widehat{\mathbf{R}} - \mathbf{R}\|_2 \le K \cdot \sqrt{\frac{s\log(d/\delta)}{n}} \lesssim 1 \qquad \text{(D.16)}$$

holds with probability at least $1 - \delta/3$, where the last inequality is valid since $n \gtrsim K^2 s \|\mathbf{R}\|_2^2 \log(d/\delta)$. Combing Eq.(D.14), Eq.(D.15) and Eq.(D.16), we have

$$\text{I} \lesssim K\|\boldsymbol{\Sigma}\|_2 \sqrt{\frac{s\log(d/\delta)}{n}} \cdot \left(1 + \max\left\{ \sqrt{\frac{r(\boldsymbol{\Sigma}) + \log(1/\delta)}{n}}, \frac{r(\boldsymbol{\Sigma}) + \log(1/\delta)}{n} \right\}\right)$$

$$\le K\|\boldsymbol{\Sigma}\|_2 \sqrt{s\frac{\log(d/\delta)}{n}} \cdot \left(1 + \sqrt{\frac{r(\boldsymbol{\Sigma}) + \log(1/\delta)}{n}} + \frac{r(\boldsymbol{\Sigma}) + \log(1/\delta)}{n}\right). \qquad \text{(D.17)}$$

For term II, we can decompose II as follows:

$$\|\mathbf{R}\left(\widehat{\boldsymbol{\Sigma}} - \boldsymbol{\Sigma}\right)\widehat{\mathbf{R}}\|_2 \le \underbrace{\|\mathbf{R}\left(\widehat{\boldsymbol{\Sigma}} - \boldsymbol{\Sigma}\right)\mathbf{R}\|_2}_{\text{II.a}} + \underbrace{\|\mathbf{R}\left(\widehat{\boldsymbol{\Sigma}} - \boldsymbol{\Sigma}\right)\left(\widehat{\mathbf{R}} - \mathbf{R}\right)\|_2}_{\text{II.b}}.$$

For term II.a, by using Lemma D.3, we have with probability at least $1 - \delta/3$,

$$\text{II.a} \lesssim \|\mathbf{R}\boldsymbol{\Sigma}\mathbf{R}\|_2 \cdot \max\left\{ \sqrt{\frac{r(\mathbf{R}\boldsymbol{\Sigma}\mathbf{R})}{n}}, \frac{r(\mathbf{R}\boldsymbol{\Sigma}\mathbf{R})}{n}, \sqrt{\frac{\log(1/\delta)}{n}}, \frac{\log(1/\delta)}{n} \right\}$$

$$\lesssim \|\mathbf{R}\boldsymbol{\Sigma}\mathbf{R}\|_2 \cdot \max\left\{ \sqrt{\frac{r(\mathbf{R}\boldsymbol{\Sigma}\mathbf{R}) + \log(1/\delta)}{n}}, \frac{r(\mathbf{R}\boldsymbol{\Sigma}\mathbf{R}) + \log(1/\delta)}{n} \right\}$$

$$\le \|\mathbf{R}\boldsymbol{\Sigma}\mathbf{R}\|_2 \cdot \left( \sqrt{\frac{r(\mathbf{R}\boldsymbol{\Sigma}\mathbf{R}) + \log(1/\delta)}{n}} + \frac{r(\mathbf{R}\boldsymbol{\Sigma}\mathbf{R}) + \log(1/\delta)}{n} \right) \qquad \text{(D.18)}$$

Similar to the proof for bounding I, we can obtain that

$$\text{II.b} \lesssim K\|\boldsymbol{\Sigma}\|_2 \sqrt{\frac{s\log(d/\delta)}{n}} \cdot \left(1 + \sqrt{\frac{r(\boldsymbol{\Sigma}) + \log(1/\delta)}{n}} + \frac{r(\boldsymbol{\Sigma}) + \log(1/\delta)}{n}\right). \qquad \text{(D.19)}$$

For term III, we have

$$\text{III} = \|\mathbf{R}\boldsymbol{\Sigma}\left(\widehat{\mathbf{R}} - \mathbf{R}\right)\|_2 \le |\mathbf{R}\|_2\|\boldsymbol{\Sigma}\|_2 K(\|\mathbf{w}^\star\|_1 + 1) \cdot \sqrt{\frac{s\log(d/\delta)}{n}}, \qquad \text{(D.20)}$$

where the last inequality is by Eq.(D.16).

Combining Eq.(D.17), Eq.(D.18), Eq.(D.19) and Eq.(D.20) and taking the union bound, we obtain that with probability at least $1 - \delta$,

$$\|\widehat{\mathbf{R}}\widehat{\mathbf{\Sigma}}\widehat{\mathbf{R}} - \mathbf{R}\mathbf{\Sigma}\mathbf{R}\|_2 \le \mathrm{I} + \mathrm{II} + \mathrm{III}$$

$$\lesssim K\|\mathbf{\Sigma}\|_2(\|\mathbf{w}^\star\|_1 + 1)\sqrt{\frac{\log{(d/\delta)}}{n}} \cdot \left(1 + \sqrt{\frac{r(\mathbf{\Sigma}) + \log{(1/\delta)}}{n}} + \frac{r(\mathbf{\Sigma}) + \log{(1/\delta)}}{n}\right)$$

$$+ \|\mathbf{R}\mathbf{\Sigma}\mathbf{R}\|_2 \cdot \left(\sqrt{\frac{r(\mathbf{R}\mathbf{\Sigma}\mathbf{R}) + \log{(1/\delta)}}{n}} + \frac{r(\mathbf{R}\mathbf{\Sigma}\mathbf{R}) + \log{(1/\delta)}}{n}\right)$$

$$+ \|\mathbf{R}\|_2\|\mathbf{\Sigma}\|_2 K(\|\mathbf{w}^\star\|_1 + 1) \cdot \sqrt{\frac{\log{(d/\delta)}}{n}}$$

$$\le (K\|\mathbf{\Sigma}\|_2(\|\mathbf{w}^\star\|_1 + 1) + \|\mathbf{R}\mathbf{\Sigma}\mathbf{R}\|_2 + \|\mathbf{R}\|_2\|\mathbf{\Sigma}\|_2 K(\|\mathbf{w}^\star\|_1 + 1))$$

$$\cdot \left(\sqrt{\frac{\log{(d/\delta)}}{n}} \cdot \left(2 + \sqrt{\frac{r(\mathbf{\Sigma}) + \log{(1/\delta)}}{n}} + \frac{r(\mathbf{\Sigma}) + \log{(1/\delta)}}{n}\right)\right.$$

$$\left. + \sqrt{\frac{r(\mathbf{R}\mathbf{\Sigma}\mathbf{R}) + \log{(1/\delta)}}{n}} + \frac{r(\mathbf{R}\mathbf{\Sigma}\mathbf{R}) + \log{(1/\delta)}}{n}\right)$$

$$\lesssim \widetilde{C}_{\mathrm{cov}} \cdot \left(\sqrt{\frac{r(\mathbf{R}\mathbf{\Sigma}\mathbf{R}) + \log{(1/\delta)}}{n}} + \frac{\sqrt{r(\mathbf{\Sigma})\log{(d/\delta)}} + r(\mathbf{R}\mathbf{\Sigma}\mathbf{R}) + \log(d/\delta)}{n}\right.$$

$$\left. + \frac{r(\mathbf{\Sigma})\sqrt{\log{(d/\delta)}} + \log^{3/2}{(d/\delta)}}{n^{3/2}}\right)$$

$$\lesssim \widetilde{C}_{\mathrm{cov}} \cdot \mathrm{poly}(\log{(d/\delta)}) \cdot \left(\sqrt{\frac{r(\mathbf{R}\mathbf{\Sigma}\mathbf{R})}{n}} + \frac{\sqrt{r(\mathbf{\Sigma})} + r(\mathbf{R}\mathbf{\Sigma}\mathbf{R})}{n} + \frac{r(\mathbf{\Sigma})}{n^{3/2}}\right),$$

where the second last inequality is by $aa' + bb' + cc' \le (a + b + c)(a' + b' + c')$ for $a, a', b, b', c, c' \ge 0$. Here $\widetilde{C}_{\mathrm{cov}} = K\|\mathbf{\Sigma}\|_2(\|\mathbf{w}^\star\|_1 + 1) + \|\mathbf{R}\mathbf{\Sigma}\mathbf{R}\|_2 + \|\mathbf{R}\|_2\|\mathbf{\Sigma}\|_2 K(\|\mathbf{w}^\star\|_1 + 1) \lesssim \sqrt{s}$. $\qquad \square$

*Proof of Lemma D.5.* By the triangle inequality, we have

$$\left\|\widehat{\mathbf{R}}\left(\mathbf{\Sigma} - \widehat{\mathbf{\Sigma}}\right)\widehat{\mathbf{R}}\right\|_2$$

$$= \left\|\mathbf{R}\left(\mathbf{\Sigma} - \widehat{\mathbf{\Sigma}}\right)\mathbf{R} + \mathbf{R}\left(\mathbf{\Sigma} - \widehat{\mathbf{\Sigma}}\right)\left(\widehat{\mathbf{R}} - \mathbf{R}\right) + \left(\widehat{\mathbf{R}} - \mathbf{R}\right)\left(\mathbf{\Sigma} - \widehat{\mathbf{\Sigma}}\right)\mathbf{R} + \left(\widehat{\mathbf{R}} - \mathbf{R}\right)\left(\mathbf{\Sigma} - \widehat{\mathbf{\Sigma}}\right)\left(\widehat{\mathbf{R}} - \mathbf{R}\right)\right\|_2$$

$$\le \left\|\mathbf{R}\left(\mathbf{\Sigma} - \widehat{\mathbf{\Sigma}}\right)\mathbf{R}\right\|_2 + \left\|\mathbf{R}\left(\mathbf{\Sigma} - \widehat{\mathbf{\Sigma}}\right)\left(\widehat{\mathbf{R}} - \mathbf{R}\right)\right\|_2 + \left\|\left(\widehat{\mathbf{R}} - \mathbf{R}\right)\left(\mathbf{\Sigma} - \widehat{\mathbf{\Sigma}}\right)\mathbf{R}\right\|_2 + \left\|\left(\widehat{\mathbf{R}} - \mathbf{R}\right)\left(\mathbf{\Sigma} - \widehat{\mathbf{\Sigma}}\right)\left(\widehat{\mathbf{R}} - \mathbf{R}\right)\right\|_2.$$

Following the proof of Lemma D.4, we can prove that with probability at least $1 - \delta$,

$$\left\|\widehat{\mathbf{R}}\left(\mathbf{\Sigma} - \widehat{\mathbf{\Sigma}}\right)\widehat{\mathbf{R}}\right\|_2 \lesssim \widetilde{\alpha}(n, \delta) \le 1/t, \tag{D.21}$$

where the last inequality is by $\mathcal{E}_2$. By Eq.(D.21), we have

$$\widehat{\mathbf{R}}\left(\mathbf{\Sigma} - \widehat{\mathbf{\Sigma}}\right)\widehat{\mathbf{R}} \preceq 1/t \cdot \mathbf{I}.$$

Hence, we obtain that

$$\mathrm{I} \lesssim \mathbf{w}^{\star\top}\overline{\mathbf{R}}\left(\mathbf{I} - \eta\widehat{\mathbf{\Sigma}}\right)^t \cdot 1/t \cdot \mathbf{I} \cdot \left(\mathbf{I} - \eta\widehat{\mathbf{\Sigma}}\right)^t\overline{\mathbf{R}}\mathbf{w}^\star$$

$$= \frac{1}{t}\mathbf{w}^{\star\top}\overline{\mathbf{R}}\left(\mathbf{I} - \eta\widehat{\mathbf{\Sigma}}\right)^{2t}\overline{\mathbf{R}}\mathbf{w}^\star$$

$$\le \frac{1}{t}\mathbf{w}^{\star\top}\overline{\mathbf{R}}\overline{\mathbf{R}}\mathbf{w}^\star \qquad\qquad \left(\text{by } \left(\mathbf{I} - \eta\widehat{\mathbf{\Sigma}}\right)^{2t} \preceq \mathbf{I}\right)$$

$$\le \frac{1}{t}\|\mathbf{w}^\star\|_2^2, \tag{D.22}$$

where the last line by $\overline{\mathbf{R}} \preceq \frac{2}{\beta} \cdot \mathbf{I}$. For the term II, we have

$$\mathrm{II} = \mathbf{w}^{\star\top}\overline{\mathbf{R}}\left(\mathbf{I} - \eta\widehat{\boldsymbol{\Sigma}}\right)^t \widehat{\mathbf{R}}\boldsymbol{\Sigma}\widehat{\mathbf{R}}\left(\mathbf{I} - \eta\widehat{\boldsymbol{\Sigma}}\right)^t \overline{\mathbf{R}}\mathbf{w}^{\star}$$

$$\lesssim \frac{1}{\eta t}\mathbf{w}^{\star\top}\overline{\mathbf{R}}\overline{\mathbf{R}}\mathbf{w}^{\star}$$

$$\frac{1}{\eta t \beta^2}\|\mathbf{w}^{\star}\|_2^2 \leq \frac{1}{\eta t \beta^2}, \tag{D.23}$$

where the second last line is by the fact that $x(1 - x)^k \leq 1/(k + 1)$ for all $x \in [0, 1]$ and all $k > 0$. $\qquad\square$

*Proof of Lemma D.6.* Similar to the proof of Lemma D.5, with probability at least $1 - \delta$, we have $\widehat{\mathbf{R}}\left(\boldsymbol{\Sigma} - \widehat{\boldsymbol{\Sigma}}\right)\widehat{\mathbf{R}} \preceq \frac{1}{t} \cdot \mathbf{I}$. Then we have

$$\mathrm{I} = \frac{\eta^2}{n^2}\epsilon^\top \mathbf{X}\widehat{\mathbf{R}} \sum_{i=1}^{t}\left(\mathbf{I} - \eta\widehat{\boldsymbol{\Sigma}}\right)^{i-1}\widehat{\mathbf{R}}\left(\boldsymbol{\Sigma} - \widehat{\boldsymbol{\Sigma}}\right)\widehat{\mathbf{R}}\sum_{i=1}^{t}\left(\mathbf{I} - \eta\widehat{\boldsymbol{\Sigma}}\right)^{i-1}\widehat{\mathbf{R}}\mathbf{X}^\top\epsilon$$

$$\lesssim \frac{\eta^2}{tn^2}\epsilon^\top \mathbf{X}\widehat{\mathbf{R}}\sum_{i=1}^{t}\left(\mathbf{I} - \eta\widehat{\boldsymbol{\Sigma}}\right)^{i-1}\sum_{i=1}^{t}\left(\mathbf{I} - \eta\widehat{\boldsymbol{\Sigma}}\right)^{i-1}\widehat{\mathbf{R}}\mathbf{X}^\top\epsilon$$

$$\leq \frac{\eta^2 t}{n^2}\epsilon^\top \mathbf{X}\widehat{\mathbf{R}}\widehat{\mathbf{R}}\mathbf{X}^\top\epsilon$$

$$= \frac{\eta^2 t}{n^2} \cdot \left\|\widehat{\mathbf{R}}\mathbf{X}^\top\epsilon\right\|_2^2,$$

where the second last line is by $\sum_{i=1}^{t}\left(\mathbf{I} - \eta\widehat{\boldsymbol{\Sigma}}\right)^{i-1} \preceq t \cdot \mathbf{I}$. By the fact that $x(1 - x)^k \leq 1/(k + 1)$ for all $x \in [0, 1]$ and all $k > 0$, we have

$$\mathrm{II} = \frac{\eta^2}{n^2}\epsilon^\top \mathbf{X}\widehat{\mathbf{R}}\sum_{i=1}^{t}\left(\mathbf{I} - \eta\widehat{\boldsymbol{\Sigma}}\right)^{i-1}\widehat{\mathbf{R}}\boldsymbol{\Sigma}\widehat{\mathbf{R}}\sum_{i=1}^{t}\left(\mathbf{I} - \eta\widehat{\boldsymbol{\Sigma}}\right)^{i-1}\widehat{\mathbf{R}}\mathbf{X}^\top\epsilon$$

$$= \frac{\eta}{n^2}\epsilon^\top \mathbf{X}\widehat{\mathbf{R}}\left(\sum_{i,j=1}^{t}\left(\mathbf{I} - \eta\widehat{\boldsymbol{\Sigma}}\right)^{i+j-2}\eta\widehat{\mathbf{R}}\widehat{\boldsymbol{\Sigma}}\right)\widehat{\mathbf{R}}\mathbf{X}^\top\epsilon$$

$$\leq \frac{\eta}{n^2} \cdot \left(\sum_{i,j=1}^{t}\frac{1}{i + j - 1}\right)\left\|\widehat{\mathbf{R}}\mathbf{X}^\top\epsilon\right\|_2^2$$

$$\leq \frac{\eta t}{n^2} \cdot \left(\sum_{i=1}^{t}\frac{1}{i}\right)\left\|\widehat{\mathbf{R}}\mathbf{X}^\top\epsilon\right\|_2^2$$

$$\lesssim \frac{\eta t \log t}{n^2} \cdot \left\|\widehat{\mathbf{R}}\mathbf{X}^\top\epsilon\right\|_2^2,$$

where the last inequality is by the fact that $\sum_{i=1}^{t}\frac{1}{i} \lesssim \log t$. $\qquad\square$

*Proof of Lemma D.7.* First, we can decompose $\left\|\frac{1}{n} \cdot \widehat{\mathbf{R}}\mathbf{X}^\top\epsilon\right\|_2^2$ by

$$\left\|\frac{1}{n} \cdot \widehat{\mathbf{R}}\mathbf{X}^\top\epsilon\right\|_2^2 \lesssim \left\|\frac{1}{n} \cdot \mathbf{R}\mathbf{X}^\top\epsilon\right\|_2^2 + \left\|\frac{1}{n} \cdot \left(\widehat{\mathbf{R}} - \mathbf{R}\right)\mathbf{X}^\top\epsilon\right\|_2^2.$$

Let $\mathbf{z}_i = \mathbf{R}\mathbf{x}_i$, then $\mathbf{z}_i \sim \mathsf{N}(\mathbf{G})$, where $\mathbf{G} := \mathbf{R}\boldsymbol{\Sigma}\mathbf{R}$. For any $i, j$, by Lemma 2.7.7 in Vershynin [46], there exists an absolute constant $C$ such that $\epsilon_j z_{ji}$ is a sub-exponential random variable with

$$\|\epsilon_j z_{ji}\|_{\Psi_1} \leq C\sigma\sqrt{G_{ii}}.$$

By applying Bernstein's inequality Vershynin [46, Theorem 2.8.1], for any $1 \leq i \leq d$, we have that

$$\left| \frac{1}{n} \sum_{j=1}^{n} \epsilon_j z_{ji} - \mathbb{E}[\epsilon_1 z_{1i}] \right| = \left| \frac{1}{n} \sum_{j=1}^{n} \epsilon_j z_{ji} \right|$$

$$\lesssim \sigma \sqrt{G_{ii}} \cdot \max \left\{ \sqrt{\frac{\log(d/\delta)}{n}}, \frac{\log(d/\delta)}{n} \right\} = \sigma \sqrt{G_{ii}} \cdot \sqrt{\frac{\log(d/\delta)}{n}} \qquad (\text{D.24})$$

hold with probability $1 - \frac{\delta}{3d}$, where the last inequality is due to $n \geq \mathcal{O}(\log(d/\delta))$. By taking the union bound, we obtain that

$$\left| \frac{1}{n} \sum_{j=1}^{n} \epsilon_j z_{ji} \right| \lesssim \sigma \sqrt{G_{ii}} \cdot \sqrt{\frac{\log(d/\delta)}{n}}$$

holds for any $i$, with probability $1 - \frac{\delta}{3}$. Then we have

$$\mathrm{I} = \sum_{i=1}^{d} \left( \frac{1}{n} \sum_{j=1}^{n} \epsilon_j \mathbf{z}_{ji} \right)^2 \lesssim \sum_{i=1}^{d} \sigma^2 G_{ii} \cdot \frac{\log(d/\delta)}{n} = \frac{\sigma^2 \mathrm{Tr}(\mathbf{R}\boldsymbol{\Sigma}\mathbf{R}) \log(d/\delta)}{n}.$$

In the same way, we can prove that with probability at least $1 - \delta/3$,

$$\left\| \frac{1}{n} \mathbf{X}^\top \epsilon \right\|_2^2 \lesssim \frac{\sigma^2 \mathrm{Tr}(\Sigma) \log(d/\delta)}{n}. \qquad (\text{D.25})$$

By applying Lemma D.1, with probability at least $1 - \delta/3$, we have

$$\left\| \widehat{\mathbf{R}} - \mathbf{R} \right\|_2^2 \lesssim \frac{s \log(d/\delta)}{n}. \qquad (\text{D.26})$$

By Eq.(D.25) and Eq.(D.26), with probability $1 - 2\delta/3$, we have

$$\left\| \frac{1}{n} \cdot \left( \widehat{\mathbf{R}} - \mathbf{R} \right) \mathbf{X}^\top \epsilon \right\|_2^2 \leq \left\| \widehat{\mathbf{R}} - \mathbf{R} \right\|_2^2 \left\| \frac{1}{n} \mathbf{X}^\top \epsilon \right\|_2^2 \lesssim \frac{\sigma^2 s \mathrm{Tr}(\boldsymbol{\Sigma}) \log^2(d/\delta)}{n^2}.$$

By taking the union bound, we derive the desired result. $\qquad \square$

## E  Proof for Theorem 5.2

To simplify the notations, we use $\mathbf{w}_t$ to denote $\mathbf{w}_{\mathrm{gd}}^t$.

**Lemma E.1.** *with probability at least $1 - \delta$, we have*

$$\left\| \widehat{\boldsymbol{\Sigma}} - \boldsymbol{\Sigma} \right\| \lesssim \alpha(n, \delta), \qquad (\text{E.1})$$

*where $\alpha(n, \delta) = \sqrt{\frac{\mathrm{Tr}(\boldsymbol{\Sigma}) + \log(1/\delta)}{n}} + \frac{\mathrm{Tr}(\boldsymbol{\Sigma}) + \log(1/\delta)}{n}$. As a result, when $n \gtrsim t^2 (\mathrm{Tr}(\boldsymbol{\Sigma}) + \log(1/\delta))$, with probability at least $1 - \delta$,*

$$\left\| \widehat{\boldsymbol{\Sigma}} - \boldsymbol{\Sigma} \right\| \lesssim 1/t.$$

*Proof of Lemma E.1.* By Lemma D.3, we have

$$\|\widehat{\boldsymbol{\Sigma}} - \boldsymbol{\Sigma}\|_2 \leq c\|\boldsymbol{\Sigma}\|_2 \cdot \max \left\{ \sqrt{\frac{r(\boldsymbol{\Sigma})}{n}}, \frac{r(\boldsymbol{\Sigma})}{n}, \sqrt{\frac{\log(1/\delta)}{n}}, \frac{\log(1/\delta)}{n} \right\}$$

$$\lesssim \max \left\{ \sqrt{\frac{r(\boldsymbol{\Sigma}) + \log(1/\delta)}{n}}, \frac{r(\boldsymbol{\Sigma}) + \log(1/\delta)}{n} \right\}$$

$$\leq \sqrt{\frac{r(\boldsymbol{\Sigma}) + \log(1/\delta)}{n}} + \frac{r(\boldsymbol{\Sigma}) + \log(1/\delta)}{n} \qquad (\text{E.2})$$

holds with probability at least $1 - \delta$, where the last line is by the inequality that $\max\{a, b\} \leq a + b$ for all $a, b \geq 0$. $\qquad \square$

We define the event $\mathcal{E}$ as follows:

$$\mathcal{E} := \Big\{ \mathbf{R\Sigma R}\|_2 \lesssim \alpha(n, \delta) \le 1/t \Big\}.$$

By Lemma E.1, $\mathbb{P}(\mathcal{E}) \ge 1 - \delta$. Hereafter, we condition on $\mathcal{E}$.

**Bias-variance Decomposition** Similar to Eq.(D.1), we have

$$\mathbf{w}_t = \Big( \mathbf{I} - \big( \mathbf{I} - \eta\widehat{\mathbf{\Sigma}} \big)^t \Big) \mathbf{w}^\star + \frac{1}{n} \sum_{i=1}^{t} \big( \mathbf{I} - \eta\widehat{\mathbf{\Sigma}} \big)^{i-1} \mathbf{X}^\top \epsilon. \tag{E.3}$$

In the same way, we can decompose the risk $\mathcal{E}(\mathbf{w}_t)$ by

$$\mathcal{E}(\mathbf{w}_t) = \underbrace{\Big\| \mathbf{\Sigma}^{1/2} \big( \mathbf{I} - \eta\widehat{\mathbf{\Sigma}} \big)^t \mathbf{w}^\star \Big\|_2^2}_{\text{Bias}} + \eta^2 \underbrace{\Big\| \mathbf{\Sigma}^{1/2} \Big( \frac{1}{n} \sum_{i=1}^{t} \big( \mathbf{I} - \eta\widehat{\mathbf{\Sigma}} \big)^{i-1} \mathbf{X}^\top \epsilon \Big) \Big\|_2^2}_{\text{Variance}}. \tag{E.4}$$

Bounding the Bias

$$\begin{aligned} \text{Bias} &= \mathbf{w}^{\star\top} \big( \mathbf{I} - \eta\widehat{\mathbf{\Sigma}} \big)^t \mathbf{\Sigma} \big( \mathbf{I} - \eta\widehat{\mathbf{\Sigma}} \big)^t \mathbf{w}^\star \\ &= \underbrace{\mathbf{w}^{\star\top} \big( \mathbf{I} - \eta\widehat{\mathbf{\Sigma}} \big)^t \big( \mathbf{\Sigma} - \widehat{\mathbf{\Sigma}} \big) \big( \mathbf{I} - \eta\widehat{\mathbf{\Sigma}} \big)^t \mathbf{w}^\star}_{\text{I}} + \underbrace{\mathbf{w}^{\star\top} \big( \mathbf{I} - \eta\widehat{\mathbf{\Sigma}} \big)^t \widehat{\mathbf{\Sigma}} \big( \mathbf{I} - \eta\widehat{\mathbf{\Sigma}} \big)^t \mathbf{w}^\star}_{\text{II}}. \end{aligned}$$

Similar to the proof of Lemma D.5, we have the following lemma.
**Lemma E.2.** *On $\mathcal{E}$, we have*

$$\text{I} \lesssim \frac{1}{t}$$

*and*

$$\text{II} \lesssim \frac{1}{\eta t}$$

*hold with probability at least $1 - \delta$.*

As a result, the bound of the bias term is given by

$$\text{Bias} \le \frac{1}{\eta t} + \frac{1}{t} \lesssim \frac{1}{\eta t}. \tag{E.5}$$

Bounding the Variance By using the same way of the proof for bounding the variance term of Theorem 5.1, we have the following lemma.
**Lemma E.3.** *On $\mathcal{E}$, with probability at least $1 - \delta$, we have that*

$$\text{Variance} \lesssim \eta t \log t \cdot \Big\| \frac{1}{n} \cdot \mathbf{X}^\top \epsilon \Big\|_2^2 \lesssim \eta t \log t \cdot \frac{\sigma^2 \text{Tr}(\Sigma) \log (d/\delta)}{n}. \tag{E.6}$$

Combining Eq.(E.5) and Eq.(E.6), we obtain that

$$\mathcal{E}(\mathbf{w}_t) \lesssim \frac{1}{\eta t} + \eta t \log t \cdot \frac{\sigma^2 \text{Tr}(\Sigma) \log (d/\delta)}{n} \lesssim \log t \cdot \sqrt{\frac{\sigma^2 \text{Tr}(\Sigma) \log (d/\delta)}{n}},$$

when $\eta t \simeq \Big( \frac{\sigma^2 \text{Tr}(\Sigma) \log (d/\delta)}{n} \Big)^{-1/2}$

# F   Proof for Theorem 5.3

To simplify the notation, we use $\widehat{\mathbf{w}}_t$ to denote $\widetilde{\mathbf{w}}_{\text{gd}}^t$ and $\mathbf{w}_t$ to denote $\mathbf{w}_{\text{gd}}^t$.

## F.1 Proof for the upper bound of the excess risk

When $\boldsymbol{\Sigma} = \mathbf{I}$, by Eq.(D.2), we have

$$\mathcal{E}(\widehat{\mathbf{w}}_t) = \underbrace{\left\| \widehat{\mathbf{R}} \left( \mathbf{I} - \eta \widehat{\boldsymbol{\Sigma}} \right)^t \overline{\mathbf{R}} \mathbf{w}^\star \right\|_2^2}_{\text{Bias}} + \underbrace{\eta^2 \left\| \widehat{\mathbf{R}} \left( \frac{1}{n} \sum_{i=1}^{t} \left( \mathbf{I} - \eta \widehat{\boldsymbol{\Sigma}} \right)^{i-1} \widetilde{\mathbf{X}}^\top \epsilon \right) \right\|_2^2}_{\text{Variance}}.$$

Following the proof of Theorem 5.1, it holds that

$$\text{Variance} \lesssim \eta t \log t \cdot \frac{\sigma^2 \log (d/\delta)}{n} + \frac{\sigma^2 s d \log^2 (d/\delta)}{n^2}$$

with probability at least $1 - \delta$, when $n \gtrsim t^2 s d^{2/3}$

Similar to the proof of Lemma D.2, we can prove that

$$\widehat{r}_i \geq \frac{r_i}{2} \;\; \forall i \in \mathcal{S}, \qquad\qquad \widehat{r}_i \lesssim 1 \;\; \forall i,$$

with probability at least $1 - \delta$.

When $\boldsymbol{\Sigma} = \mathbf{I}$, by Lemma D.4, we have that

$$\left\| \widehat{\mathbf{R}} \widehat{\boldsymbol{\Sigma}} \widehat{\mathbf{R}} - \mathbf{R} \boldsymbol{\Sigma} \mathbf{R} \right\|_2 \lesssim \frac{\beta^2}{t}$$

holds with probability at least $1 - \delta$, when $n \gtrsim \frac{t^2 \|\mathbf{w}^\star\|_1^2 d^{2/3}}{\beta^4}$. As a result, $\mathbf{R} \boldsymbol{\Sigma} \mathbf{R} - \frac{\beta^2}{t} \cdot \mathbf{I} \preceq \widehat{\mathbf{R}} \widehat{\boldsymbol{\Sigma}} \widehat{\mathbf{R}}$. Hereafter, we condition on the above events. For the bias term, we have

$$\left\| \widehat{\mathbf{R}} \left( \mathbf{I} - \eta \widehat{\boldsymbol{\Sigma}} \right)^t \overline{\mathbf{R}} \mathbf{w}^\star \right\|_2^2 \leq \left\| \widehat{\mathbf{R}} \right\|_2^2 \cdot \left\| \left( \mathbf{I} - \eta \widehat{\boldsymbol{\Sigma}} \right)^t \overline{\mathbf{R}} \mathbf{w}^\star \right\|_2^2$$

$$\leq \mathbf{w}^{\star\top} \mathbf{R} \left( \mathbf{I} - \eta \widehat{\boldsymbol{\Sigma}} \right)^{2t} \overline{\mathbf{R}} \mathbf{w}^\star$$

$$\lesssim \mathbf{w}^{\star\top} \mathbf{R} \left( \mathbf{I} - \eta \left( \mathbf{R} \boldsymbol{\Sigma} \mathbf{R} - \frac{\beta^2}{t} \cdot \mathbf{I} \right) \right)^{2t} \overline{\mathbf{R}} \mathbf{w}^\star$$

$$= \sum_{i \in \mathcal{S}} (w_i^\star / \widehat{r}_i)^2 \cdot \left( 1 - \eta \left( (w_i^\star)^2 - \frac{\beta^2}{t} \right) \right)^{2t}$$

$$\leq s \cdot \left( 1 - \eta \beta^2 / 2 \right)^{2t},$$

where the last line is by the definition of $\beta$. When $t \gtrsim \log \left( \frac{\sigma^2}{ns} \right) / \left( 2 \log \left( 1 - \eta \beta^2 / 2 \right) \right)$, we have

$$\text{Bias} = \left\| \widehat{\mathbf{R}} \left( \mathbf{I} - \eta \widehat{\boldsymbol{\Sigma}} \right)^t \overline{\mathbf{R}} \mathbf{w}^\star \right\|_2^2 \leq \frac{\sigma^2}{n}. \tag{F.1}$$

When $\eta \beta^2 / 2 \leq 1/2$, there exist a $c > 0$, such that

$$\log \left( 1 - \eta \beta^2 / 2 \right) \geq c \eta \beta^2 / 2.$$

Hence, the variance term is bounded by

$$\text{Variance} \lesssim \eta t \log t \cdot \left( \frac{\sigma^2 \log (d/\delta)}{n} + \frac{\sigma^2 \|\mathbf{w}^\star\|_1^2 d \log^2 (d/\delta)}{n^2} \right)$$

$$\lesssim \frac{\sigma^2 \log^2 (ns/\sigma^2) \log^2 (d/\delta)}{\beta^2} \cdot \left( \frac{s}{n} + \frac{ds}{n^2} \right), \tag{F.2}$$

where the last line is by $\|\mathbf{w}^\star\|_1 \leq s \cdot \|\mathbf{w}^\star\|_2^2 = s$ and $\eta t \lesssim \frac{\log (ns/\sigma^2)}{\beta^2}$. Combining Eq.(F.1) and Eq.(F.2), we have that

$$\mathcal{E}(\widehat{\mathbf{w}}_t) \lesssim \frac{\sigma^2}{n} + \frac{\sigma^2 \log^2 (ns/\sigma^2) \log^2 (d/\delta)}{\beta^2} \cdot \left( \frac{1}{n} + \frac{ds}{n^2} \right) \lesssim \frac{\sigma^2 \log^2 (ns/\sigma^2) \log^2 (d/\delta)}{\beta^2} \cdot \left( \frac{1}{n} + \frac{ds}{n^2} \right),$$

when $n \gtrsim \frac{t^2 s d^{2/3}}{\beta^4} \geq \frac{t^2 \|\mathbf{w}^\star\|_1^2 d^{2/3}}{\beta^4}$ and $t \gtrsim \frac{\log(ns)}{\eta \beta^2}$. When $w_i^\star \in \mathsf{U}\{-1/\sqrt{s}, 1/\sqrt{s}\}$, $\beta = 1/\sqrt{s}$. In this case, we have that

$$\mathcal{E}(\widehat{\mathbf{w}}_t) \lesssim \sigma^2 \log^2\left(ns/\sigma^2\right) \log^2\left(d/\delta\right) \cdot \left(\frac{s}{n} + \frac{ds^2}{n^2}\right),$$

when $n \gtrsim t^2 s^3 d^{2/3}$ and $t \gtrsim \frac{\log(ns)}{\eta s}$.

## F.2    Lower bound for Ridge Regression

When $n \gtrsim d + \log(1/\delta)$, by Lemma D.3, we have that $\frac{1}{2} \cdot \mathbf{I} \preceq \widehat{\boldsymbol{\Sigma}} \preceq 2 \cdot \mathbf{I}$ For the ridge estimator $\widehat{\mathbf{w}}_\lambda = \frac{1}{n} \cdot \left(\widehat{\boldsymbol{\Sigma}} + \lambda \cdot \mathbf{I}\right)^{-1} \mathbf{X}^\top \mathbf{y}$, we have

$$\mathbb{E}_{\mathbf{w}^\star}[\mathcal{E}(\widehat{\mathbf{w}}_\lambda)] = \left\|\left(\mathbf{I} - \left(\widehat{\boldsymbol{\Sigma}} + \lambda \mathbf{I}\right)^{-1} \widehat{\boldsymbol{\Sigma}}\right) \mathbf{w}^\star\right\|_2^2 + \left\|\frac{1}{n} \cdot \left(\widehat{\boldsymbol{\Sigma}} + \lambda \cdot \mathbf{I}\right)^{-1} \mathbf{X}^\top \epsilon\right\|_2^2$$

$$\geq \left\|\frac{1}{n} \cdot \left(\widehat{\boldsymbol{\Sigma}} + \lambda \cdot \mathbf{I}\right)^{-1} \mathbf{X}^\top \epsilon\right\|_2^2.$$

By Lemma D.3, when $\frac{1}{2} \cdot \mathbf{I} \preceq \widehat{\boldsymbol{\Sigma}} \preceq 2 \cdot \mathbf{I}$, with probability at least $1 - \delta$, we have

$$\mathbb{E}_{\mathbf{w}^\star}[\mathcal{E}(\widehat{\mathbf{w}}_\lambda)] \geq \left\|\frac{1}{n} \cdot \left(\widehat{\boldsymbol{\Sigma}} + \lambda \cdot \mathbf{I}\right)^{-1} \mathbf{X}^\top \epsilon\right\|_2^2$$

$$= \frac{1}{n^2} \cdot \epsilon^\top \mathbf{X} \left(\widehat{\boldsymbol{\Sigma}} + \lambda \mathbf{I}\right)^{-2} \mathbf{X}^\top \epsilon$$

$$\geq \frac{1}{n^2 (2 + \lambda)^2} \cdot \epsilon^\top \mathbf{X} \mathbf{X}^\top \epsilon,$$

where the last line is due to the fact that $\widehat{\boldsymbol{\Sigma}} + \lambda \mathbf{I} \preceq (2 + \lambda) \cdot \mathbf{I}$.

**Lemma F.1.** *Given $X$ such that $\frac{1}{2}\mathbf{I} \preceq \widehat{\boldsymbol{\Sigma}} \preceq 2\mathbf{I}$, it holds that*

$$\left\|\frac{1}{n} \mathbf{X}^\top \epsilon\right\|_2^2 \gtrsim \frac{\sigma^2 d}{n},$$

*with probability at least $1 - \delta$, when $n \geq \mathcal{O}(\log(1/\delta))$.*

*Proof of Lemma F.1.* We consider the singular value decomposition of $\frac{1}{\sqrt{n}} \mathbf{X}^\top$: $\frac{1}{\sqrt{n}} \mathbf{X}^\top = \mathbf{U} \boldsymbol{\Lambda} \mathbf{V}^\top$, where $\mathbf{U} \in \mathbb{R}^{d \times d}$ is an orthogonal matrix, $\boldsymbol{\Lambda} \in \mathbb{R}^{d \times n}$ is a rectangular diagonal matrix with non-negative real numbers on the diagonal, $\mathbf{V} \in \mathbb{R}^{n \times n}$ is an orthogonal matrix. Let $\{\sigma_1, \ldots, \sigma_d\}$ be the singular values of $\frac{1}{\sqrt{n}} \mathbf{X}^\top$. Then we have

$$\left\|\frac{1}{n} \mathbf{X}^\top \epsilon\right\|_2^2 = \left\|\frac{1}{\sqrt{n}} \mathbf{U} \boldsymbol{\Lambda} \mathbf{V}^\top \epsilon\right\|_2^2 = \left\|\frac{1}{\sqrt{n}} \boldsymbol{\Lambda} \mathbf{V}^\top \epsilon\right\|_2^2$$

$$= \left\|\frac{1}{\sqrt{n}} \boldsymbol{\Lambda} \widetilde{\epsilon}\right\|_2^2 = \frac{1}{n} \sum_{i=1}^d \sigma_i^2 \widetilde{\epsilon}_i^2,$$

where $\widetilde{\epsilon} = \mathbf{V}^\top \epsilon \sim \mathsf{N}(\mathbf{0}, \mathbf{I})$. By ][Lemma 22], we have

$$\left|\left\|\frac{1}{n} \mathbf{X}^\top \epsilon\right\|_2^2 - \mathbb{E}\left[\left\|\frac{1}{n} \mathbf{X}^\top \epsilon\right\|_2^2\right]\right| \lesssim \sigma^2 \max\left\{\frac{\sqrt{\sum_{i=1}^d \sigma_i^4 \log(1/\delta)}}{n}, \frac{\max_i \sigma_i^2 \log(1/\delta)}{n}\right\}$$

$$\lesssim \sigma^2 \max\left\{\frac{\sqrt{d \log(1/\delta)}}{n}, \frac{\log(1/\delta)}{n}\right\}, \tag{F.3}$$

where the last line is valid since $\{\sigma_1^2, \ldots, \sigma_d^2\}$ is the eigenvalues of $\widehat{\boldsymbol{\Sigma}} = \frac{1}{n}\mathbf{X}^\top\mathbf{X}$ and $\frac{1}{2}\mathbf{I} \preceq \widehat{\boldsymbol{\Sigma}} \preceq 2\mathbf{I}$. By Eq.(F.3), we obtain that

$$\left\|\frac{1}{n}\mathbf{X}^\top\epsilon\right\|_2^2 \geq \mathbb{E}\left[\left\|\frac{1}{n}\mathbf{X}^\top\epsilon\right\|_2^2\right] - \sigma^2 \max\left\{\frac{\sqrt{d\log(1/\delta)}}{n}, \frac{\log(1/\delta)}{n}\right\}$$

$$= \sigma^2 \sum_{i=1}^d \sigma_i^2 - \sigma^2 \max\left\{\frac{\sqrt{d\log(1/\delta)}}{n}, \frac{\log(1/\delta)}{n}\right\}$$

$$= \sigma^2 \frac{d}{n} - \sigma^2 \max\left\{\frac{\sqrt{d\log(1/\delta)}}{n}, \frac{\log(1/\delta)}{n}\right\} \qquad \text{(by } \tfrac{1}{2}\mathbf{I} \preceq \widehat{\boldsymbol{\Sigma}} \preceq 2\mathbf{I}\text{)}$$

$$\lesssim \sigma^2 \frac{d}{n},$$

where the last line is due to $d \geq \mathcal{O}(\log(1/\delta))$. $\qquad\qquad\square$

Next, we define the event $\mathcal{E}$ as follows:

$$\mathcal{E}_{\text{ridge}} := \left\{\frac{1}{2}\mathbf{I} \preceq \widehat{\boldsymbol{\Sigma}} \preceq 2\mathbf{I}, \left\|\frac{1}{n}\mathbf{X}^\top\epsilon\right\|_2^2 \gtrsim \frac{\sigma^2 d}{n}\right\}.$$

By Lemma F.1, we have $\mathbb{P}(\mathcal{E}) \geq 1 - \delta$ when $n \geq \mathcal{O}(d) \geq \mathcal{O}(\log(1/\delta))$. On $\mathcal{E}_{\text{ridge}}$, we have

$$\mathbb{E}_{\mathbf{w}^\star}[\mathcal{E}(\widehat{\mathbf{w}}_\lambda)] \gtrsim \frac{\sigma^2 d}{(1+\lambda)^2 n}. \tag{F.4}$$

When $d \gtrsim n + \log(1/\delta)$, by Lemma D.3, with probability at least $1 - \delta$, we have that $\frac{d}{2} \cdot \mathbf{I} \preceq \mathbf{X}\mathbf{X}^\top \preceq 2d \cdot \mathbf{I}$. Hereafter, we condition on this event. By direct calculation, we can decompose the excess risk by

$$\mathbb{E}_{\mathbf{w}^\star}[\mathcal{E}(\widehat{\mathbf{w}}_\lambda)] = \mathbb{E}_{\mathbf{w}^\star}\left\|\left(\mathbf{I} - \left(\widehat{\boldsymbol{\Sigma}} + \lambda\mathbf{I}\right)^{-1}\widehat{\boldsymbol{\Sigma}}\right)\mathbf{w}^\star\right\|_2^2 + \left\|\frac{1}{n} \cdot \left(\widehat{\boldsymbol{\Sigma}} + \lambda \cdot \mathbf{I}\right)^{-1}\mathbf{X}^\top\epsilon\right\|_2^2.$$

For the first term, we have

$$\mathbb{E}_{\mathbf{w}^\star}\left\|\left(\mathbf{I} - \left(\widehat{\boldsymbol{\Sigma}} + \lambda\mathbf{I}\right)^{-1}\widehat{\boldsymbol{\Sigma}}\right)\mathbf{w}^\star\right\|_2^2 = \mathbb{E}_{\mathbf{w}^\star}\left\|\left(\mathbf{I} - \mathbf{X}^\top\left(\mathbf{X}\mathbf{X}^\top + n\lambda\mathbf{I}\right)^{-1}\mathbf{X}\right)\mathbf{w}^\star\right\|_2^2$$

$$= (1 - \frac{n}{d})\mathbb{E}_{\mathbf{w}^\star}\left[\|\mathbf{w}^\star\|_2^2\right], \tag{F.5}$$

$$= 1 - \frac{n}{d} \tag{F.6}$$

where the last line is due to $\left(\mathbf{I} - \mathbf{X}^\top\left(\mathbf{X}\mathbf{X}^\top + n\lambda\mathbf{I}\right)^{-1}\mathbf{X}\right)$ is a $d - n$ space.

$$\left\|\frac{1}{n} \cdot \left(\widehat{\boldsymbol{\Sigma}} + \lambda \cdot \mathbf{I}\right)^{-1}\mathbf{X}^\top\epsilon\right\|_2^2 = \epsilon^\top\mathbf{X}\mathbf{X}^\top\left(\mathbf{X}\mathbf{X}^\top + n\lambda\mathbf{I}\right)^{-2}\epsilon$$

$$\geq \frac{dn}{2(2d + n\lambda)^2} \cdot \frac{1}{n}\sum_{i=1}^n \epsilon_i^2, \tag{F.7}$$

where the first line is by $\left(\mathbf{X}^\top\mathbf{X} + n\lambda\mathbf{I}\right)^{-1}\mathbf{X}^\top = \mathbf{X}^\top\left(\mathbf{X}\mathbf{X}^\top + n\lambda\mathbf{I}\right)^{-1}$ and the last line is by $\frac{d}{2(d+n\lambda)^2} \cdot \mathbf{I} \preceq \mathbf{X}\mathbf{X}^\top\left(\mathbf{X}\mathbf{X}^\top + n\lambda\mathbf{I}\right)^{-2}$. By Tsigler and Bartlett [44, Lemma 22], we obatain that

$$\left|\sum_{i=1}^n \epsilon_i^2 - n\sigma^2\right| \lesssim \sigma^2\sqrt{n\log(1/\delta)} + \sigma^2$$

holds with probability at least $1 - \delta$. When $n \gtrsim \log(1/\delta)$, we have $\left|\sum_{i=1}^n \epsilon_i^2 - n\sigma^2\right| \geq \frac{n\sigma^2}{2}$ holds with probability at least $1 - \delta$. Taking the union bound, we obtain that

$$\mathbb{E}_{\mathbf{w}^\star}[\mathcal{E}(\widehat{\mathbf{w}}_\lambda)] \gtrsim 1 - \frac{n}{d} + \sigma^2 \cdot \frac{dn}{2(2d + n\lambda)^2} \gtrsim 1 - \frac{n}{d} + \sigma^2\frac{n}{(1+\lambda)^2 d}. \tag{F.8}$$

### F.3 Lower Bound for Finite-Step GD

We first consider the case where $n \gtrsim d + \log(1/\delta)$. Define the event $\mathcal{E}_{\mathrm{GD}}$ by $\mathcal{E}_{\mathrm{GD}} = \left\{ \frac{1}{2} \cdot \mathbf{I} \preceq \widehat{\boldsymbol{\Sigma}} \preceq 2\mathbf{I} \right\}$. By Lemma D.3, $\mathbb{P}(\mathcal{E}_{\mathrm{GD}}) \geq 1 - \delta$. By Eq.(E.4), we have

$$
\begin{aligned}
\mathbb{E}_{\mathbf{w}^\star}[\mathcal{E}(\mathbf{w}_t)] = \mathbb{E}_{\mathbf{w}^\star} & \left\| \left(\mathbf{I} - \eta\widehat{\boldsymbol{\Sigma}}\right)^t \mathbf{w}^\star \right\|_2^2 + \eta^2 \left\| \left( \frac{1}{n}\sum_{i=1}^{t} \left(\mathbf{I} - \eta\widehat{\boldsymbol{\Sigma}}\right)^{i-1} \mathbf{X}^\top \epsilon \right) \right\|_2^2 \\
& \geq \eta \left\| \left( \frac{1}{n}\sum_{i=1}^{t} \left(\mathbf{I} - \eta\widehat{\boldsymbol{\Sigma}}\right)^{i-1} \mathbf{X}^\top \epsilon \right) \right\|_2^2 \\
& = \frac{\eta^2}{n^2} \cdot \left\| \left( \widehat{\boldsymbol{\Sigma}} \left( \mathbf{I} - \left(\mathbf{I} - \eta\widehat{\boldsymbol{\Sigma}}\right)^t \right)^{-1} \right)^{-1} \mathbf{X}^\top \epsilon \right\|_2^2 \\
& \gtrsim \frac{\eta^2}{n^2} \cdot \left\| \left( \widehat{\boldsymbol{\Sigma}} + \frac{1}{\eta t} \cdot \mathbf{I} \right)^{-1} \mathbf{X}^\top \epsilon \right\|_2^2 \\
& \gtrsim \sigma^2 \frac{\eta^2 d}{(1 + 1/(\eta t))^2 n},
\end{aligned}
$$

where the second last line is by $\widehat{\boldsymbol{\Sigma}} \left( \mathbf{I} - \left(\mathbf{I} - \eta\widehat{\boldsymbol{\Sigma}}\right)^t \right)^{-1} \preceq \boldsymbol{\Sigma} + \frac{2}{t\eta} \cdot \mathbf{I}$ and the last line is by Eq.(F.4).

We then consider the case where $d \gtrsim n + \log(1/\delta)$. Define the event $\mathcal{E}'_{\mathrm{GD}} = \left\{ \frac{d}{2} \cdot \mathbf{I} \preceq \mathbf{X}\mathbf{X}^\top \prec 2d\mathbf{I} \right\}$. By Lemma D.3, $\mathbb{P}(\mathcal{E}'_{\mathrm{GD}}) \geq 1 - \delta$. Following the proof of Zou et al. [58, Theorem 4.3], we have

$$
\begin{aligned}
\mathbb{E}_{\mathbf{w}^\star}[\mathcal{E}(\mathbf{w}_t)] \geq \mathbb{E}_{\mathbf{w}^\star} & \left\| \left( \mathbf{I} - \mathbf{X}^\top \left( \mathbf{X}\mathbf{X}^\top + \frac{n}{\eta t}\mathbf{I} \right)^{-1} \mathbf{X} \right) \right\|_2^2 + \left\| \frac{1}{n}\mathbf{X}^\top \left( \mathbf{X}\mathbf{X}^\top + \frac{n}{\eta t}\mathbf{I} \right)^{-1} \epsilon \right\|_2^2 \\
& = 1 - \frac{n}{d} + \frac{\sigma^2 n}{\left(1 + \frac{1}{\eta t}\right)^2 d},
\end{aligned}
$$

where we use the results from Appendix F.2.

### F.4 Lower bound of OLS

Let $\mathbf{w}_{\mathrm{ols}}$ be the OLS estimator. It is easy to see $\mathbf{w}_{\mathrm{ols}} = \mathbf{w}_0$. Hence, we have

$$
\mathbb{E}_{\mathbf{w}^\star}[\mathcal{E}(\mathbf{w}_{\mathrm{ols}})] \gtrsim \begin{cases} \frac{\sigma^2 d}{n} & n \gtrsim d + \log(1/\delta) \\ 1 - \frac{n}{d} + \frac{\sigma^2 n}{d} & d \gtrsim n + \log(1/\delta), \end{cases}
$$

holds with probability at least $1 - \delta$.

## G  Additional Experiments

Here, we provide additional experiments on the decoder-only architecture and train models with different settings.

**Training Decoder-Only Transformer**  In this experiment, we adapt the same input setting and training objective as in [20]. During training, we set $n = 24$ and $k = 8$ in Eq.(G.1) (where in $y_i$, we use zero padding to align with $\mathbf{x}_i$), $d_{\mathsf{hid}} = 256$. We choose $h = 8$ and $l \in \{4, 5, 6\}$.[3] We then conduct heads assessment experiments on the trained decoder-only transformers with 10 in-context

---

[3]We also tried other settings with fewer heads or layers, but even with delicate hyperparameter tuning, decoder-only transformers with fewer heads or layers consistently failed to learn how to solve our sparse linear regression problem. A possible reason is that decoder-only transformers first need to learn the causal structure [35] and then apply an optimization algorithm to the in-context entries, which is more challenging than our encoder-based settings.

examples, as in the previous settings. The result is shown in Figure 9. We can observe that the decoder-only transformer exhibits the similar weight distribution for each layer as the encoder-based models, indicating that our algorithm may extend to decoder-only based models.

$$\mathbf{E} = \begin{pmatrix} \mathbf{x}_1 & y_1 & \mathbf{x}_2 & y_2 & \dots & \mathbf{x}_n & y_n \end{pmatrix}, \quad L = \sum_{i=k}^{n} (\widehat{y}_i - y_i)^2. \tag{G.1}$$

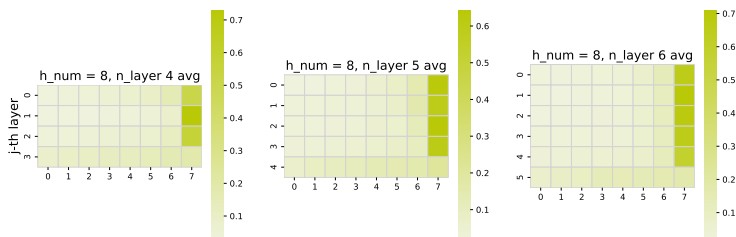

Figure 9: Heads Assessment for decoder-only transformers

**Training Models with $s = d = 16$** Here, we adapt the encoder-only transformer and the same settings as introduced in B, but set $s = d = 16$. We observe that in these cases, there is no distinct performance difference between models with different numbers of heads. As shown in Figure 3, when we set $s = 4, d = 16$, transformers with more heads ($h = 4, 8$) always perform better than models with fewer heads ($h = 1, 2$). However, in Figure 10, such a difference is unclear, which aligns well with the theoretical analysis. When $s$ is close to $d$, a clear better upper bound guarantee, as ensured in cases where $s \ll d$ may not hold.

**Training Models with non-orthogonal design** To further demonstrate the applicability of our experimental results to more general non-orthogonal settings, we conducted additional experiments by modifying the distribution of $\mathbf{x}$ to $\mathsf{N}(\mathbf{0}, \Sigma)$, where $\Sigma = \mathbf{I} + \zeta \mathbf{S}$, and $\mathbf{S}$ is a matrix of ones. We varied $\zeta$ across the values $[0, 0.1, 0.2, 0.4]$ to validate our findings. The results are presented in Figure 11, which reveals patterns similar to those observed in orthogonal design settings.

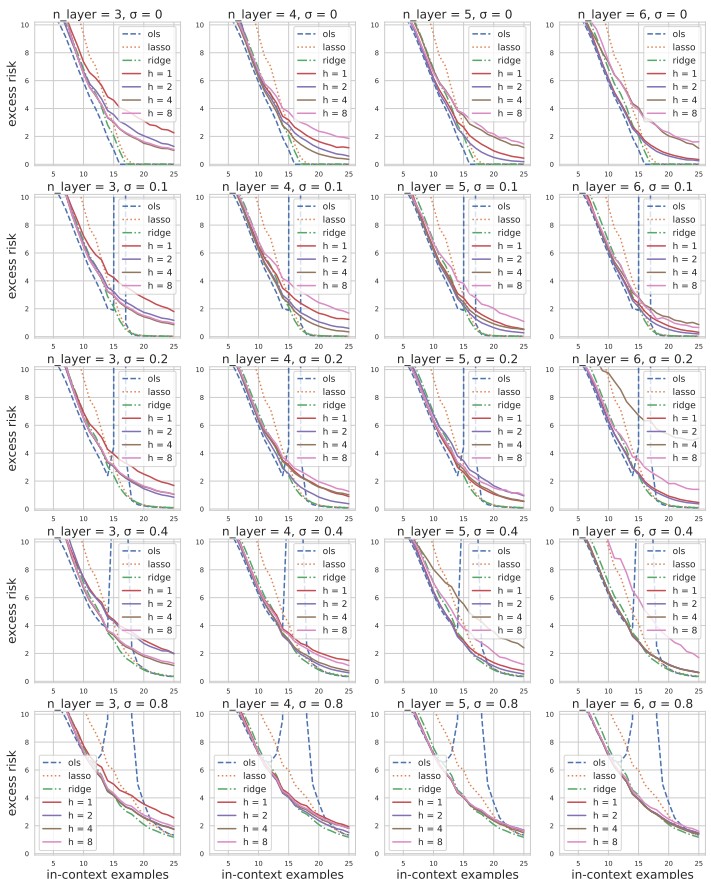

Figure 10: Train Models with $s = d = 16$

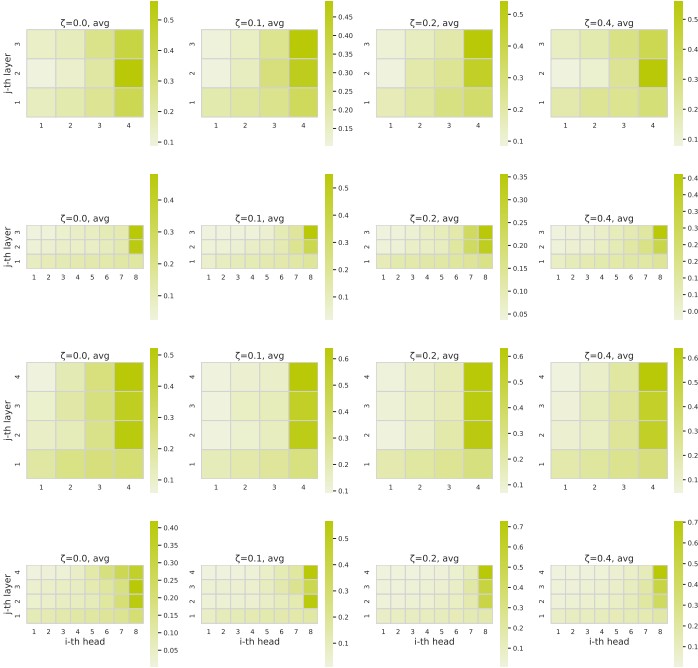

Figure 11: Train Models with $\mathbf{x} \sim \mathsf{N}(\mathbf{0}, \Sigma)$, where $\Sigma = \mathbf{I} + \zeta\mathbf{S}$.

