# OpenReview forum: "How Transformers Utilize Multi-Head Attention in In-Context Learning? A Case Study on Sparse Linear Regression"
_NeurIPS.cc/2024/Conference — NeurIPS 2024 poster_

### Official Review · Reviewer_Bazq · 2024-06-27

**Soundness:** 3
**Presentation:** 3
**Contribution:** 2
**Rating:** 5
**Confidence:** 4

**Summary:**

This paper empirically studies how the different heads works across different layers of the transformer: while the first layer uses all heads, the later layers mainly relies on a single head. In addition, the authors also propose a preprocess-then-optimize algorithm.

**Strengths:**

This paper is easy to understand and provides interesting theoretical results.

**Weaknesses:**

The main flow of Section 4 and 5 is that, it (1) constructs a first layer which can perform the exact preprocessing considered in the proposed algorith, (2) studies the generalization of the preprocessing, (3) constructs a the later layers which can perform gradient descent, and finally (4) studies the generalization of the optimization. The paper uses a linear attention-only transformer.

Given the above, there are several main weaknesses regarding the contributions in this paper:

(1) Missing theory or theoretical intuition to explain why multi-head attention works in the way observed in this paper. Given the series of in-context learning research which focuses on aligning multi-layer transformers with gradient descent, the contribution is limited.

(2) The linear attention-only transformer can be improved to better align with the practical LLMs. In the recent year, various studies consider single/multi-head attention with softmax attention [1,2,3]. The authors are encouraged to add softmax attention in the paper. In particular, compared to the linear attention result in [4], the softmax result in [1,2,3] shows that softmax attention works differently from linear attention. The authors are also encouraged to study how MLP works in the system. These elements in the transformer may result in different behavior compared to the linear attention-only transformer.

(3) The paper constructs a transformer which aligns with the preprocess-then-optimize algorithm, rather than showing that a trained multi-layer transformer indeed works in the way designed in this paper. The authors need to provide more evidences for this, e.g., [5]. The authors may also want to highlight their different to [5]. A convergence analysis similar to [1] and [2] is even better.

(4) The generalization properties are built upon a linear model, whose analysis is supposed to be routine. The authors may also highlight the challenges when performing the analysis.

Given the above limitation, the theoretical contributions are not sufficient enough. There is no critical difference between existing theories in literature and the study in this paper.

References:

[1] Huang, Yu, Yuan Cheng, and Yingbin Liang. "In-context convergence of transformers." arXiv preprint arXiv:2310.05249 (2023).

[2] Chen, Siyu, et al. "Training dynamics of multi-head softmax attention for in-context learning: Emergence, convergence, and optimality." arXiv preprint arXiv:2402.19442 (2024).

[3] Cui, Yingqian, et al. "Superiority of multi-head attention in in-context linear regression." arXiv preprint arXiv:2401.17426 (2024).

[4] Zhang, Ruiqi, Spencer Frei, and Peter L. Bartlett. "Trained transformers learn linear models in-context." arXiv preprint arXiv:2306.09927 (2023).

[5] Ahn, Kwangjun, et al. "Transformers learn to implement preconditioned gradient descent for in-context learning." Advances in Neural Information Processing Systems 36 (2024).

**Questions:**

I would like to raise my score if the authors could provide an extra detailed theoretical result for any one of the above weaknesses. The more the better. Thanks.

---

> ### Author Rebuttal · Authors · 2024-08-07
>
> We sincerely appreciate the time and effort you've invested in reviewing our work. We've addressed your questions and concerns as follows:
>
> > 1. *The difference between our work and other works towards aligning multi-layer transformers with gradient descent, like [3,4,5].*
>
> Thank you for your insightful reference paper, we will add more discussion about them in our revised version. We want to highlight that our work is not just focused on the expressive power of transformers (such as their ability to implement novel optimization algorithms). Instead, we aim to answer the question: "How do trained transformers tend to use multi-head attention for in-context learning?" We first gain insights from trained transformers (in Section 3) and then propose an algorithm to explain our observations (Sections 4 and 5). While [4,5] focus on single-head attention and [3] explores the expressive power of single-layer transformers, our work seeks to understand the hidden mechanisms of trained multi-layer transformers through a novel perspective. We believe our findings are both interesting and extensible to more real-world tasks and complex models, warranting further investigation in future work.
>
> > 2. *Theoretical intuition for why multi-head attention works in the way observed in this paper, and the linking between softmax attention and linear attention*.
>
> An empirical intuition for the benefit of multi-head attention in sparse linear regression is that it enables the transformer to process different subspaces of the input data using separate attention heads, as discussed in Section 4.1. Here we can provide a more theoretical explanation for the working mechanism of multi-head attention in sparse linear regression, building on [2]. Consider I = d/k, representing I different tasks, each focusing on a sparse linear regression problem with k assigned dimensions, and $g_i$ is the same for different tasks i (i.e. the task can be seen as a combination of I sparse linear regression problem). Based on the results for ms-attn in [2] (eq 4.3 & theorem 4.2), the optimal solution involves each attention head processing a specific subspace while zeroing out other dimensions, aligning with our Proposition 4.1.
>
> Assuming sufficiently large $d_i, d$ and $L$ (assumptions 2.1 and 2.2 in [2]), the parameter for $w_{qk}^h$ is scaled by $1/\sqrt{d_i}$, allowing softmax attention to converge to linear attention:$x̂_{ij} \approx\frac{\sqrt{d_i}}{N} \sum_{k = 1}^{n} (\frac{\langle x_{kj} ,x_{ij}\rangle}{\sqrt{d_i}} + 1) y_{k}=\langle\frac1N \sum{x_{kj} y_k}, x_{ij}\rangle + \frac{\sqrt{d_i}}N \sum{y_k} $
>
> Here, we omit the term $\frac1{1 + e d_i \phi_i L^{-1} }$  for $u_i$ (eq4.3 in [2]), as it's constant across i in our setting. The left term $\frac1N \sum{x_{kj} y_k}$ can be interpreted as our preprocess coefficient $\hat{r}_i$ (defined in eq4.1 in our paper), while $\frac{\sqrt{d_i}}N \sum{y_k}$ remains constant across different tasks i.
>
> While our results for the first layer's multi-head attention resemble those in [2], we emphasize two key differences: 1) we elucidate the role of multi-head attention in multi-layer transformers through a new preprocessing perspective, and 2) we provide both theoretical and empirical explanations for the advantages of multi-head attention, demonstrating a potential integration mechanism between the first layer and subsequent layers. Although the convergence behavior of deeper layers remains an open question, our theoretical intuition above suggests a potential approach: first assuming that subsequent layers optimize data through gradient descent, then prove that the first layer converges to utilizing multi-head attention for data preprocessing, we believe this warrants further investigation in future research.
>
>
> > 3. The role of MLP in our setting.
>
> In our linear setting, learning from context with linear attention is widely adopted in theoretical analyses. While linear attention incorporating MLP layers can implement complex algorithms, it's challenging for a trained transformer to adopt the exact solution we've constructed. In the attached PDF, we compare the performance of linear attention-only transformers with and without MLP layers, demonstrating that the difference between these models is not significant, thus justifying our simplification.
>
> > 4. Challenges in our analysis on the linear model
>
> Although we are analyzing a linear model, when we analyze the convergence of Preprocessing+GD, we encounter some non-trivial challenges. We point out two main challenges encountered in analyzing this problem as follows: 1) In general linear models, data are assumed to be independent, but in our setting, the data preprocessing involves the utilization of all training data points, making the preprocessed data become non-independent. This raises a great challenge in proving the concentration-related results. 2) In the analysis for the preprocessed data, the preprocessing matrix is not commutable with either the empirical data covariance matrix or the population data covariance matrix, thus many prior techniques in the literature cannot be applied and we need to develop new techniques accordingly. This also raises many challenges in obtaining reasonable risk bounds in our analysis.
>
> ---
>
> Finally, we want to emphasize that large language models are intricate systems, and the theoretical understanding of transformers remains in its early stages. Similar to early physics investigations, our approach of gaining insights from carefully designed experiments and then explaining our observations is a reasonable path toward this goal. While we acknowledge numerous areas for future research, we believe our observations and results are both interesting and meaningful, contributing to our understanding of the mechanisms behind large language models, both empirically and theoretically.

---

> > ### Comment · Reviewer_Bazq · 2024-08-07
> >
> > I appreciate the authors in addressing my comments. I have raised my score from 4 to 5.

---

### Official Review · Reviewer_Swt3 · 2024-07-04

**Soundness:** 3
**Presentation:** 4
**Contribution:** 3
**Rating:** 7
**Confidence:** 3

**Summary:**

This work presents an analysis of how Transformers perform in-context learning by experimenting with a sparse linear regression problem setup. The author’s analysis combines both empirical and theoretical analysis. They first examine the properties of real models using pruning and probing methods. From this, they identify a key property of the Transformers, that multiple heads are needed in the first layer but in the remaining layers only a single attention head is necessary. From this, they present theoretical analysis and propose a preprocess-then-optimize learning approach to explain the observed behavior in transformers. The authors also present results showing the effectiveness of this learning approach.

**Strengths:**

While this work is not in my area of expertise, I believe the work is well organized and clearly presented. I appreciate the authors combination of empirical and theoretical analysis, to demonstrate that the proposed learning method may actually be employed by real transformers in real tasks and are not only achievable through hand-crafted weights. The authors results are mathematically rigorous as well in supporting the proposed preprocess-then-optimize method.

**Weaknesses:**

The main weakness I see in the work pertains to the simplifications made for the analysis. The authors also acknowledge these limitations in section 7. In summary, the analysis focuses on a simplified problem (linear regression) and uses a simplified version of a transformer, which is also done by other similar works. As a result, it is hard to say if a similar learning approach is learned by transformers in more practical real-world problems. It is also impossible to say how the removed layers may alter the learning process in a full transformer. Overall, I believe this work warrants an accept rating, though I will defer to the other reviewers if they have significant issues with the work.

**Questions:**

Line 108: should this instead read "to perform a tractable theoretical investigation"?
Line 209: typo "proprocessing" instead of "preprocessing"

**Limitations:**

The authors have discussed the limitations I mentioned in the section above. I do not see any clear potential negative societal impact for this work.

---

> ### Author Rebuttal · Authors · 2024-08-07
>
> Thank you for your insightful comments! We sincerely appreciate the time and effort you've dedicated to providing thoughtful reviews. We've addressed your concerns as follows:
>
> > 1. *The theoretical analysis is based on a simplified transformer maybe hard to apply the results to more practical real-world problems*.
>
> While our analysis uses a simplified transformer model, we believe this widely adopted simplification captures the essence of the model and provides valuable insights applicable to real-world transformers. In the attached pdf, we use additional experiments to demonstrate that MLP layers are less critical in our sparse linear regression setting, and our theoretical and experimental results can be extended to various data distributions. Moreover, based on the recent studies about the parameter distribution of real word transformers, our results are promising to be extended to more practical settings, for example [1] and [2] demonstrate through experiments that parameters in deeper layers are less critical compared to those in shallower layers, additionally, [3] shows that multi-head attention sublayers exhibit low-rank structure in large language models. We believe our findings are not limited to simple sparse linear regression settings but can be extended to more complex ICL and other real-world tasks, and this phenomenon is worth further investigation in future work.
>
> \[1]Gromov, A., Tirumala, K., Shapourian, H., Glorioso, P., & Roberts, D. A. (2024). The unreasonable ineffectiveness of the deeper layers. arXiv preprint arXiv:2403.17887.
>
> \[2]Men, X., Xu, M., Zhang, Q., Wang, B., Lin, H., Lu, Y., ... & Chen, W. (2024). Shortgpt: Layers in large language models are more redundant than you expect. arXiv preprint arXiv:2403.03853.
>
> \[3] Li, G., Tang, Y., & Zhang, W. (2024). LoRAP: Transformer Sub-Layers Deserve Differentiated Structured Compression for Large Language Models. ICML2024
>
> > 2. *Line 108: should this instead read "to perform a tractable theoretical investigation"? Line 209: typo "proprocessing" instead of "preprocessing".*
>
> Thank you for pointing out our typos and sorry for any confusion they may have caused, the "intractable" in Line 108 should be "tractable" and "proprocessing" should be "preprocessing", we will correct these typos in our revised version.

---

> > ### Comment · Reviewer_Swt3 · 2024-08-12
> >
> > I thank the authors for their responses and additional experiments. I will increase my rating from 6 to 7 as I support accepting this work.

---

### Official Review · Reviewer_uBeS · 2024-07-12

**Soundness:** 3
**Presentation:** 4
**Contribution:** 3
**Rating:** 7
**Confidence:** 4

**Summary:**

This paper studies the mechanism of transformers under the in-context sparse linear regression problem. The authors reveal that the transformer pre-trained for this task has the first layer preprocessing the data, and the remaining layers implement gradient descent. More intriguingly, only one head in the second to last layers is dominantly utilized.

**Strengths:**

1. The paper is a nice extension of the recent line of work on ICL formulated by regression problems. The study of multi-head also provides an important direction.
2. The theory is solid and clear.
3. The authors have found many interesting phenomena in their setting. I especially find their study on multi-head very interesting and beneficial for LLM interpretability.

**Weaknesses:**

1. The P-probing lacks a controlled experiment. Should you also try regressing on the hidden states before the first layer? I understand that $h=1$ might be a controlled experiment, but it may still have some preprocessing on the $x$.
2. Is Theorems 5.1 and 5.2 a fair comparison? Should the $\tilde{w}^t_{gd}$ in Theorem 5.1 also include the parameters in the first preprocessing layer? My understanding is that we want to compare the excess risk of "preprocessing+gd" v.s. " gd". Now we are comparing "gd with preprocessed data" and "gd".
3. The orthogonal design is an over-simplified setting.

**Questions:**

1. I find the observation that only one head dominates in subsequent layers very interesting. Does this also occur in other settings of ICL tasks?

**Limitations:**

The orthogonal design is an over-simplified setting, which the authors do not properly address.
1. It is necessary to include ablation studies with non-orthogonal design. It's fine to get different results without the orthogonality, but it would be important to report the findings.
2. If the MLP layer is added, an alternative lasso method should be available that uses the closed-form solution under orthogonal design. Can the authors present the experiment results?
Overall, I think there should be more ablation studies in different settings. Although a simplified setting is fine for theory, I want to see experimental results in more realistic settings.

---

> ### Author Rebuttal · Authors · 2024-08-07
>
> We sincerely appreciate the time and effort you spent on thoughtful reviews and comments. We address your comments below:
>
> > **Q1**: *The P-probing lacks a controlled experiment. Should you also try regressing on the hidden states before the first layer? I understand that $h=1$ might be a controlled experiment, but it may still have some preprocessing on the  $x$.*
>
> Thank you for your insightful suggestions. In our preprocessing algorithm, the transformer employs multiple attention heads ($h>1$ ) to process each subspace differently, thereby enhancing the optimization steps for subsequent layers in our sparse linear regression task. However, a single-head transformer ($h=1$ ) can only process the entire space uniformly, lacking the ability to differentiate between subspaces. Consequently, single-head transformers have very limited "preprocessing ability," so we believe using $h=1$ serves as a reasonable "no preprocessing" comparison.
>
> We appreciate your suggestion to use hidden states without attention module for preprocessing as another point of comparison. We have provided additional results in the attached PDF.
>
> > **Q2**:*Is Theorems 5.1 and 5.2 a fair comparison? Should the $\tilde{w}^t_{gd}$ in Theorem 5.1 also include the parameters in the first preprocessing layer? My understanding is that we want to compare the excess risk of "preprocessing+gd" v.s. " gd". Now we are comparing "gd with preprocessed data" and "gd".*
>
> Sorry for causing the misunderstanding.  We would like to clarify that the comparison between Theorem 5.1 and 5.2 are made in terms of two “mechanisms” (corresponding to multi-head transformer and single-head transformer), which we believe is fair and the model parameters do not need to be included. In particular, based on Propositions 4.1 and 4.2, we conjecture that the working mechanism of multi-head transformer is “preprocess-then-optimize”, i.e., performing GD on the preprocessed data, while the working mechanism of the single-head transformer is conjectured to perform the “purely-optimize mechanism” on the original data [1]. When applying our results to $L$ layers transformers, $t$-step gd with preprocessed data ($L = t+1$) truly requires just one more layer than  $t$-step gd ($L = t$), However, factor $L$ only appears in the log term of our bound (Theorem 5.1&5.2), hence it doesn’t affect our results (which are interpreted in orders).
>
> To this end, it is fair to compare GD with preprocessed data and GD to demonstrate the superiority of our preprocess-then-optimize mechanism. We will make this clear in the revised version.
>
> [1]Ahn, K., Cheng, X., Daneshmand, H., & Sra, S. (2024).Transformers learn to implement preconditioned gradient descent for in-context learning. NeurIPS 2024
>
> > **Q3**: *I find the observation that only one head dominates in subsequent layers very interesting. Does this also occur in other settings of ICL tasks?.*
>
> Thank you for your interest. We believe a similar phenomenon occurs in other settings of ICL tasks as well, and transformers may utilize similar preprocessing-then-optimize algorithms across different layers. While this is somewhat beyond the scope of our paper, we can gain insights from experimental results of other works. For instance, recent studies by [2] and [3] demonstrate through experiments that parameters in deeper layers are less critical compared to those in shallower layers. Additionally, [4] shows that multi-head attention sublayers exhibit low-rank structure in large language models. We believe our findings are not limited to simple sparse linear regression settings but can be extended to more complex ICL and other real-world tasks, and such phenomenon is worth further investigation in future works.
>
> [2]Gromov, A., Tirumala, K., Shapourian, P., & Roberts, D.A.The unreasonable ineffectiveness of the deeper layers.arXiv preprint arXiv:2403.17887.
>
> [3]Men, X., Xu, M., Zhang, Q., ... & Chen, W.Shortgpt: Layers in large language models are more redundant than you expect.arXiv preprint arXiv:2403.03853.
>
> [4] Li, G., Tang, Y., & Zhang, W. LoRAP:Transformer Sub-Layers Deserve Differentiated Structured Compression for Large Language Models.ICML2024
>
> > **Q4** *The orthogonal design is an over-simplified setting, which the authors do not properly address.*
> > > **Q4.1**: *It is necessary to include ablation studies with non-orthogonal design. It's fine to get different results without the orthogonality, but it would be important to report the findings.*
> > > **Q4.2**: *If the MLP layer is added, an alternative lasso method should be available that uses the closed-form solution under orthogonal design. Can the authors present the experiment results?*
>
> [**A4.1**]: Thank you for your suggestions. Our experimental and theoretical results can indeed be extended to non-orthogonal designs. To validate this, we conducted additional experiments by modifying the distribution of x to x∼N(0,Σ), where Σ=I+ζS, and S is a matrix filled with 1. We varied ζ across [0,0.1,0.2,0.4] to further verify our findings. The results, which are consistent with those presented in Sections 3 and 6, can be found in the attached PDF. We will incorporate a more detailed discussion of these non-orthogonal design experiments in our revised manuscript.
>
> [**A4.2**]: While it's true that there exist specially designed transformers capable of solving the lasso problem using a closed-form solution, we find that the trained transformers are more likely to use other algorithms. Here we compared the performance of linear attention transformers with and without MLP layers in the attached PDF, which shows that the inclusion of MLP layers does not significantly impact the results for this particular problem. Although different initializations may affect the results, we believe these results are sufficient to demonstrate that our choice to use a simplified attention-only transformer for theoretical analysis is reasonable and can provide valuable insights for real-world models.

---

> > ### Comment · Reviewer_uBeS · 2024-08-10
> >
> > Thanks for the authors' responses. I would like to raise my score to 7.

---

### Official Review · Reviewer_fd5H · 2024-07-12

**Soundness:** 3
**Presentation:** 3
**Contribution:** 3
**Rating:** 5
**Confidence:** 3

**Summary:**

This work seeks to provide a deeper exploration of the use of multi-heads, at different layers in a Transformer, to perform in-context
learning tasks. More specifically, the goal of the paper is to experimentally discover additional insights on the interactions of multi-headed attention across layers. Subsequently, the authors find that multiple heads are primarily used in the first layer, whilst the remaining layers of the transformer typically leverage a single head. Furthermore, the authors provide a hypothesis for their observation before building a preprocess-then-optimize algorithm.

**Strengths:**

Strengths:
- This work provides a timely analysis of the underlying mechanism of multi-headed attention in Transformer models.
- The authors provide strong empirical and theoretical evidence to justify the insights derived from on in-context learning.
- Additionally, the authors provide a novel Preprocess-then-optimize Algorithm for training Transformers.

**Weaknesses:**

Weaknesses:
- As discussed by the authors, the reviewer's primary concern with this work is the lack of account for the remainder of the Transformer architecture. In particular, the role of feed-forward layers is sidesteped when considering the effects of the Preprocess-then-optimize Algorithm.

**Questions:**

The reviewer would like to better understand how the authors think about feed-forward layers with respect to the observations made by the authors regarding the necessity of multi-headed attention in the first few layers. In particular, would alterations to the structure of the first few feedforward layers provide a beneficial impact for transformers?

**Limitations:**

The authors have clearly stated any limitations with their current work. Additionally, the reviewer does not foresee any potential negative societal impact from this paper.

---

> ### Author Rebuttal · Authors · 2024-08-07
>
> We sincerely appreciate the time and effort you spent on thoughtful reviews. We address your comments below:
>
> > **Q1**: *The reviewer would like to better understand how the authors think about feed-forward layers with respect to the observations made by the authors regarding the necessity of multi-headed attention in the first few layers. In particular, would alterations to the structure of the first few feedforward layers provide a beneficial impact for transformers?*
>
> Thank you for your insightful questions about the feed-forward layers. In transformers, we can observe that (ignoring the layer norm) the attention mechanism is the only module capable of linking tokens across different positions, while the FFN layer performs a token-wise (nonlinear) mapping without considering contextual information. For learning from context, especially in our linear sparse regression settings where contextual information is crucial for transformers to identify appropriate subspaces for sparse linear regression problems, additional (nonlinear) mapping for each token is less important. In our theoretical analysis, we primarily focus on the attention-only transformer without the FFN layer, focusing on the role of multi-head attention, such simplification is widely adopted in theoretical analyses, as seen in [1], [2], and [3]. To demonstrate the limited role of the FFN layer in our specific setting, here we provide comparative results of the linear attention model with and without MLP layers in the attached PDF.
>
> While the FFN layer's role is limited in our setting, we acknowledge its potential beneficial impact when combined with the attention layer for more complex nonlinear or real-world tasks. Transformers may utilize similar preprocessing-then-optimize algorithms across different layers. Although this is somewhat beyond the scope of our paper, insights from other works provide valuable perspectives. For instance, [4] demonstrates that transformers tend to first learn appropriate representations using MLP layers before learning from context through the attention layer. Additionally, [5] and [6] use experiments to show that parameters in deeper layers are less critical compared to those in shallower layers. We believe our findings are both interesting and warrant further investigation in future works.
>
> \[1]Von Oswald, J., Niklasson, E., Randazzo, E., Sacramento, J., Mordvintsev, A., Zhmoginov, A., & Vladymyrov, M. (2023, July). Transformers learn in-context by gradient descent. ICML2023
>
> \[2]Ahn, K., Cheng, X., Song, M., Yun, C., Jadbabaie, A., & Sra, S. (2023). Linear attention is (maybe) all you need (to understand transformer optimization). ICLR2024
>
> \[3]Ahn, K., Cheng, X., Daneshmand, H., & Sra, S. (2024). Transformers learn to implement preconditioned gradient descent for in-context learning. NeurIPS 2023
>
> \[4]Guo, T., Hu, W., Mei, S., Wang, H., Xiong, C., Savarese, S., & Bai, Y. (2023). How do transformers learn in-context beyond simple functions? a case study on learning with representations. ICLR2024
>
> \[5]Gromov, A., Tirumala, K., Shapourian, H., Glorioso, P., & Roberts, D. A. (2024). The unreasonable ineffectiveness of the deeper layers. arXiv preprint arXiv:2403.17887.
>
> \[6]Men, X., Xu, M., Zhang, Q., Wang, B., Lin, H., Lu, Y., ... & Chen, W. (2024). Shortgpt: Layers in large language models are more redundant than you expect. arXiv preprint arXiv:2403.03853.

---

### Author Rebuttal · Authors · 2024-08-07

We sincerely appreciate the thoughtful reviews and comments provided by all reviewers. Below, we address the main points raised, details can be found in corresponding blocks for each reviewer:

- Reviewer fd5H focused on the role of MLP layers in our setting and their potential benefits for other tasks. In our  sparse linear regression setting, nonlinear tokenwise operations are less critical. While recent works suggest MLP layers may benefit other nonlinear tasks and real-world tasks; here we provide additional experiments comparing linear transformers with and without MLP layers to demonstrate that our theoretical analysis without MLP layers is reasonable.

- Reviewer uBeS primarily questioned the experimental details and results when extending our setting to different distributions. We clarify that our experimental and theoretical results can be extended to various data distributions and provide additional experiments to support this claim.

- Reviewer Swt3 inquired about the extensibility of our experimental findings and theoretical results to other settings. We reference recent experimental results on large language models to demonstrate the potential for our findings to be applied to more realistic scenarios.

- Reviewer Bazq acknowledged our experimental findings but expressed interest in
extending our theoretical analysis to other aspects, such as training dynamics and more complex problem settings like softmax attention. We highlight that our focus is on how trained transformers utilize multi-head attention for in-context learning, and these aspects are a bit beyond the scope of this paper. To address the reviewer's concern, we provide a theoretical explanation showing that the working mechanism we proposed for the multi-head first layer is similar to the theoretical analysis for softmax multi-head attentions. We also highlight the differences between our paper and other works. We believe our findings are both interesting and extensible to more real-world tasks and complex models, warranting further investigation in future work.

Additional experiments in the attached pdf:

Fig 1 compares the performance between linear attention transformers with and without MLP layers, where MLP is initialized with Xavier initialization. We chose the performance with 10 context examples. Although there exist some delicately designed transformers that can conduct more complicated algorithms for lasso, the trained transformer tends to use other solutions. The difference between models with and without MLP layers is minimal in our linear setting, justifying our simplification for theoretical analysis for sparse linear regression problem.

Fig 2 addresses Reviewer uBeS's interest in experimental results with different data distributions. We conducted additional experiments by modifying the distribution of x to x ∼ N(0,Σ), where Σ = I + ζS, and S is a matrix filled with 1. The results show that under different data distributions, our experimental findings remain consistent with orthogonal settings (Fig. 2a). Our pre-gd algorithm still outperforms gd, and our theoretical analysis can be extended to different data distribution cases (Fig 2b, 2c).

Fig 3 responds to Reviewer uBeS's interest in p-probing results using H₀. While we believe our experimental choice of H₁ for different heads is reasonable, using H₀ is could also be another option. Note that under the transformation with the first layer, the data distribution is shifted. Consequently, it is expected that the result using H₀ differs from that using H₁, h = 1. Nevertheless, the results still demonstrate that multi-head attention can achieve lower risk, justifying our preprocessing algorithm for multi-head attention.

---

### Decision · Program_Chairs · 2024-09-25

**Decision:**

Accept (poster)

**Comment:**

This paper presents an interesting analysis of how transformers utilize multi-head attention for in-context learning, focusing on a sparse linear regression task. The key findings are that multiple attention heads are utilized in the first layer for data preprocessing, while subsequent layers primarily use a single head for optimization, supported by theoretical and emipirical explanations. All reviewers agree that the submission contributes significantly, and the comments have been addressed after the rebuttal. The AC concurs with the decision to accept the submission.